# ImagenWorld: Stress-Testing Image Generation Models with Explainable Human Evaluation on Open-ended Real-World Tasks

[†‡]Samin Mahdizadeh Sani[*♠], [†‡]Max Ku[*♠♡], Nima Jamali[♠], Matina Mahdizadeh Sani[♠], Paria Khoshtab[◇], Wei-Chieh Sun[♡], Parnian Fazel[¶], Zhi Rui Tam[♡], Thomas Chong[♡], Edisy Kin Wai Chan[♡], Donald Wai Tong Tsang[♡], Chiao-Wei Hsu[♡], Ting Wai Lam[♡], Ho Yin Sam Ng[♡], Chiafeng Chu[♡], Chak-Wing Mak[◇], Keming Wu[◇], Hiu Tung Wong[♡], Yik Chun Ho[♡], Chi Ruan[♠], Zhuofeng Li[◇], I-Sheng Fang[♡], Shih-Ying Yeh[♣♡], Ho Kei Cheng[♡§], [†]Ping Nie[◇], [‡]Wenhu Chen[♠]

[♠]University of Waterloo    [♡]G-G-G    [♣]Comfy Org    [§]University of Illinois Urbana-Champaign
[◇]Independent    [¶]Imperial College London

https://tiger-ai-lab.github.io/ImagenWorld/

## Abstract

Advances in diffusion, autoregressive, and hybrid models have enabled high-quality image synthesis for tasks such as text-to-image, editing, and reference-guided composition. Yet, existing benchmarks remain limited, either focus on isolated tasks, cover only narrow domains, or provide opaque scores without explaining failure modes. We introduce **ImagenWorld**, a benchmark of 3.6K condition sets spanning six core tasks (generation and editing, with single or multiple references) and six topical domains (artworks, photorealistic images, information graphics, textual graphics, computer graphics, and screenshots). The benchmark is supported by 20K fine-grained human annotations and an explainable evaluation schema that tags localized object-level and segment-level errors, complementing automated VLM-based metrics. Our large-scale evaluation of 14 models yields several insights: (1) models typically struggle more in editing tasks than in generation tasks, especially in local edits. (2) models excel in artistic and photorealistic settings but struggle with symbolic and text-heavy domains such as screenshots and information graphics. (3) closed-source systems lead overall, while targeted data curation (e.g., Qwen-Image) narrows the gap in text-heavy cases. (4) modern VLM-based metrics achieve Kendall accuracies up to 0.79, approximating human ranking, but fall short of fine-grained, explainable error attribution. ImagenWorld provides both a rigorous benchmark and a diagnostic tool to advance robust image generation.

## 1 Introduction

The rapid progress in generative image modeling, powered by diffusion (Rombach et al., 2022; Lipman et al., 2023), autoregressive (AR) (Yu et al., 2022; Tian et al., 2024), and hybrid architectures (OpenAI, 2025), has enabled systems capable of producing high-quality images under diverse conditioning inputs. More recent work has begun to push toward broader functionality, developing models that can handle multiple tasks—such as generation and editing—within a single framework (Deng et al., 2025; Wu et al., 2025b; Chen et al., 2025a; Google, 2025), with early evidence of real-world applicability (Chen et al., 2025a). However, evaluation has not kept pace with this modeling progress. Existing benchmarks are fragmented, often restricted to isolated tasks (e.g., text-to-image (Saharia et al., 2022; Yu et al., 2022), editing (Huang et al., 2023), or personalization (Peng et al., 2025; Li et al., 2023)) or biased toward narrow domains such as artworks (Ku et al., 2024b) or textual graphics (Tuo et al., 2024). As a result, **it remains unclear how well these unified models generalize across the full spectrum of real-world use cases.** To address this gap, we introduce

---

[*]Equal contribution
[†]Project Lead
[‡]✉ samin.mahdizadeh@gmail.com; {m3ku, wenhu.chen}@uwaterloo.ca

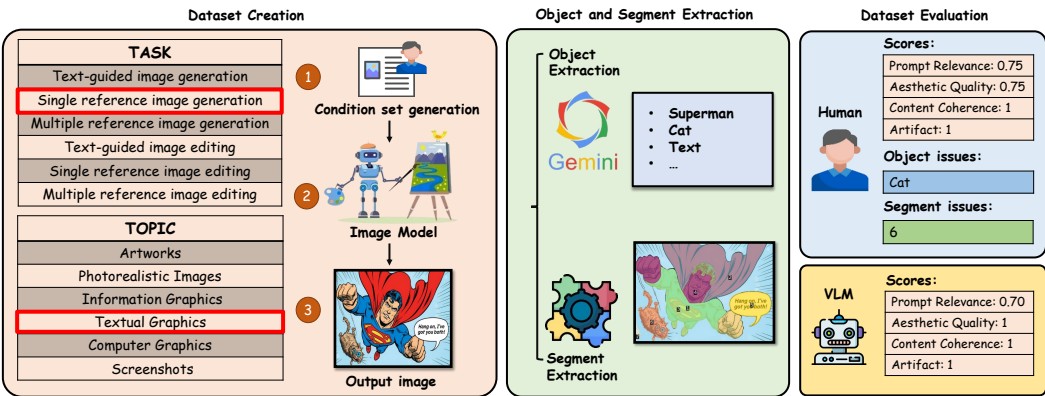

Figure 1: Overview of our dataset and evaluation pipeline, covering six content categories and six generation/editing tasks. Model outputs are assessed by both human annotators (explainable schema) and vision–language models (scores only).

**ImagenWorld**, a large-scale, human-centric benchmark comprising 3.6K condition sets designed to systematically stress-test generative models. ImagenWorld unifies six representative task types and six topical domains, creating a diverse testbed that mirrors the breadth of real-world image generation and editing. At its core, ImagenWorld relies on structured human evaluation, where annotators not only provide scores but also tag specific failure modes with textual descriptions and localized masks as illustrated in Figure 1. This schema yields explainable outcomes, revealing why models fail. To complement human judgments, we also include VLM-as-a-judge metrics, enabling comparison between human and automatic evaluators. Together, this design supports both rigorous benchmarking and forward-looking exploration of how evaluation protocols can scale. By evaluating a broad set of model families under a single protocol, ImagenWorld provides the most comprehensive picture to date of model performance and failure patterns across real-world generation and editing tasks. Our study covers 14 models in total, including 4 recent unified models capable of both generation and editing, and 10 task-specific models that serve as auxiliary baselines.

**We uncover four key insights**: (1) For editing tasks, we identify two distinct failure modes: (i) regenerating an entirely new image, and (ii) returning the input unchanged. Strikingly, models tend to exhibit one mode far more frequently than the other, suggesting a systematic bias in how they interpret editing instructions. This highlights a deeper limitation in current architectures: they lack fine-grained control mechanisms to modify localized regions without either overhauling or ignoring the input. (2) All models struggle with text-related tasks such as information graphics, screenshots, and textual graphics. However, our results reveal an exception: Qwen-Image consistently outperforms other models on textual graphics. Notably, Qwen-Image employs a synthetic data curation pipeline explicitly tailored for text-heavy images, suggesting that targeted data augmentation may be a practical path to closing this gap. This highlights that the challenge is not purely architectural, but also fundamentally tied to data design. (3) While closed-source models consistently achieve strong results across tasks, open-source models are primarily competitive in text-to-image generation, where abundant training data and community optimization have driven rapid progress. Their weaker performance in editing and multimodal composition highlights the need for further research and targeted data curation in these areas, beyond scale alone. (4) Beyond model performance, we find that modern VLM metrics achieve Kendall accuracies up to 0.79, closely matching or even exceeding human–human agreement. This suggests that modern VLMs as a judge are reliable as scalable evaluators for relative ranking in our context, but they fall short in the explainable paradigm, where humans remain indispensable for fine-grained tagging of specific failure modes.

Our contributions are threefold: (1) we introduce ImagenWorld, a diverse benchmark that unifies six core tasks and six topical domains, enabling consistent cross-model and cross-task evaluation of generative image systems; (2) we conduct the first human study of its kind, looking into the failure modes, offering new insights and observed patterns. (3) we propose a schema for explainable human evaluation, labeling object-level errors and segment-level errors to provide fine-grained, interpretable error attribution beyond scalar scores. By combining task diversity, model breadth, and diagnostic evaluation depth, ImagenWorld establishes a unified and human-centric study to record our process towards the full control of creation and manipulation in images.

## 2 RELATED WORKS

**Progress in Conditional and Multimodal Image Synthesis.** The introduction of Latent Diffusion Models (LDMs) (Rombach et al., 2022) marked a turning point, leading to a flourishing ecosystem of conditional image synthesis systems (runwayml, 2023; stability.ai, 2023) spanning diverse tasks such as instruction-driven editing (Brooks et al., 2023a; Huang et al., 2025), structural control (Zhang & Agrawala, 2023), and personalization (Ruiz et al., 2023; Yeh et al., 2024; Hu et al., 2024). While diffusion remains the dominant paradigm, alternative architectures are rapidly advancing. Autoregressive approaches (Yu et al., 2022; Tian et al., 2024) improve compositional reasoning and fidelity (Xiong et al., 2025), flow-matching models (BlackForestLabs et al., 2025) leverage ODE-native properties for potentially faster sampling, and hybrid designs such as autoregressive LLMs with diffusion decoders (Wu et al., 2024; OpenAI, 2025; Google, 2025) integrate native image generation into conversational agents. Together, these families define the current landscape of multimodal conditional image synthesis, though their evaluation remains fragmented across tasks and settings. Our work takes these developments into account by systematically studying their strengths and weaknesses under a unified evaluation framework.

**Image Synthesis Assessments and Benchmarks.** Traditional evaluations of generative image models have relied on metrics such as FID (Heusel et al., 2017) and LPIPS (Zhang et al., 2018) for image fidelity, or CLIPScore (Hessel et al., 2021) for text–image alignment. More recent approaches, including VIEScore and VQAScore (Cho et al., 2023; Hu et al., 2023; Ku et al., 2024a; Lin et al., 2024; Niu et al., 2025), use vision language models (VLM) to better capture semantic relevance, though they introduce biases and often depend on proprietary models. Human preference–driven metrics such as Pick-a-Pic (Kirstain et al., 2023a), ImageReward (Xu et al., 2023), and HPS (Ma et al., 2025) emphasize aesthetics and subjective preferences. Beyond individual metrics, benchmarks like DrawBench (Saharia et al., 2022) and PartiPrompts (Yu et al., 2022) target text-to-image fidelity, while others focus on editing (Huang et al., 2023) or personalization (Peng et al., 2025; Li et al., 2023). More recent efforts, including ImagenHub (Ku et al., 2024b) and MMIG-Bench (Hua et al., 2025), extend beyond single tasks by covering multiple generation settings and integrating both automatic and human evaluation. Gecko (Wiles et al., 2025) further scales this direction by introducing a large evaluation suite that measures text-to-image alignment across diverse human annotation templates and scoring setups. Open platforms like GenAI-Arena (Jiang et al., 2024) provide Elo-style rankings, but suffer from topic bias in user-submitted prompts. Overall, existing protocols remain task-specific or opaque, limiting the interpretability and scalability. Beyond simply adding another dataset, our work offers a new perspective: a unified benchmark across tasks and domains, complemented by structured, explainable human evaluation that can also serve as a foundation for future VLM-based automatic evaluators. Table 1 reflects how ImagenWorld differs from prior works.

Table 1: Comparison with different evaluation benchmarks on different properties.

| Method | Generation & Editing | Single Ref Guided | Multi Ref Guided | Human Rating | Topic Variety | Explainable Trace |
|---|---|---|---|---|---|---|
| ImagenHub (Ku et al., 2024b) | ✓ | ✓ | ✗ | ✓ | ✓ | ✗ |
| GenAI-Arena (Jiang et al., 2024) | ✓ | ✗ | ✗ | ✓ | ✗ | ✗ |
| DreamBench++ (Peng et al., 2025) | ✗ | ✓ | ✗ | ✗ | ✓ | ✗ |
| I²EBench (Ma et al., 2024) | ✗ | ✗ | ✗ | ✓ | ✓ | ✗ |
| ICE-Bench (Pan et al., 2025) | ✓ | ✓ | ✗ | ✗ | ✓ | ✗ |
| MMIG-Bench (Hua et al., 2025) | ✗ | ✓ | ✗ | ✓ | ✓ | ✓ |
| Rich-HF (Liang et al., 2024) | ✗ | ✗ | ✗ | ✓ | ✓ | ✓ |
| **ImagenWorld (Ours)** | ✓ | ✓ | ✓ | ✓ | ✓ | ✓ |

## 3 THE IMAGENWORLD BENCHMARK

**Problem Formulation**. To capture practical usage scenarios, we unify multiple generation and editing tasks under a common instruction-driven framework: every task is conditioned on a natural language instruction $t_{ins}$, optionally accompanied by auxiliary inputs such as a source image $I_{src}$ or a set of reference images $\mathbf{I}_R$. This unification reflects how real users typically interact with generative systems by providing instructions that guide the system to either create new images or edit existing ones. Building on this formulation, we categorize tasks into two groups: instruction-driven generation,

| Task | Text Instruction | Source Image | Image Conditions | Success Case | Failure Case |
|------|------------------|--------------|------------------|--------------|--------------|
| **TIG (CG)** | Forest scene showing bounding boxes around animals such as deer, rabbits, and birds. | – | – | | |
| **SRIG (P)** | CT scan of a brain using the reference mask as the anatomical shape. | – | | | |
| **MRIG (S)** | Generate a pet simulation game interface with pet character designs from the first and second images and the home decoration style from the third image. | – | | | |
| **TIE (T)** | Change top text to "If AI learns from humans" and the bottom text to "Who's coaching the coach?" | | – | | |
| **SRIE (A)** | Re-color the stylized portrait in image 1 using exclusively the specific color palette presented in image 2. | | | | |
| **MRIE (I)** | In step 2, update the added pasta to resemble spaghetti noodles as shown in image 2. For steps 1 and 2, refine the water graphic, depicting it with the wave pattern style found in image 3. | | | | |

Figure 2: Illustrative examples from our dataset, showing successful and failure cases for each task

where the system synthesizes a new image without a source image, and instruction-driven editing, where the system modifies an existing source image $I_{\text{src}}$ while following the instruction. Formally:

- Text-guided Image Generation (TIG): Given an instruction $t_{\text{ins}}$ in natural language, the model synthesizes a new image $y = f(t_{\text{ins}})$.

- Single Reference Image Generation (SRIG): Given an instruction $t_{\text{ins}}$ and a reference image $I_{\text{ref}}$, the model generates a new image $y = f(I_{\text{ref}}, t_{\text{ins}})$ of the referenced entity (e.g., subject, object, or layout) in a different context, pose, or environment.

- Multiple Reference Image Generation (MRIG): Given an instruction $t_{\text{ins}}$ and a set of reference images $\mathbf{I}_R = \{I_{\text{ref}}^1, I_{\text{ref}}^2, \ldots\}$, the model synthesizes a new image $y = f(\mathbf{I}_R, t_{\text{ins}})$ that composes multiple visual concepts from the instruction and references.

- Text-guided Image Editing (TIE): Given an instruction $t_{\text{ins}}$ and a source image $I_{\text{src}}$, the model produces $y = f(t_{\text{ins}}, I_{\text{src}})$ by modifying $I_{\text{src}}$ according to the instruction while preserving its core structure.

- Single Reference Image Editing (SRIE): Given an instruction $t_{\text{ins}}$, a source image $I_{\text{src}}$, and a reference image $I_{\text{ref}}$, the model edits $I_{\text{src}}$ to $y = f(I_{\text{ref}}, t_{\text{ins}}, I_{\text{src}})$, adapting the reference entity to the specified instruction or style.

- Multiple Reference Image Editing (MRIE): Given an instruction $t_{\text{ins}}$, a source image $I_{\text{src}}$, and a set of reference images $\mathbf{I}_R$, the model edits $I_{\text{src}}$ to $y = f(\mathbf{I}_R, t_{\text{ins}}, I_{\text{src}})$, aligning it with the visual attributes or semantics suggested by the instruction and references.

**Dataset Curation Pipeline.** To construct ImagenWorld, we curated a large-scale dataset through a combination of human annotation and automated refinement. Annotators wrote natural language prompts and paired them with corresponding reference or source images, ensuring that each instance aligned with one of the six benchmark tasks. To reflect real-world applications, our dataset covers six major topics: **Artworks (A), Photorealistic Images (P), Information Graphics (I), Textual Graphics (T), Computer Graphics (CG)**, and **Screenshots (S)**, each further divided into fine-grained subtopics to guarantee diverse use cases. In total, our dataset contains 3.6K entries, with 100 samples for each task–topic combination. Figure 2 shows representative examples from our dataset (see Appendix A.6 for details).

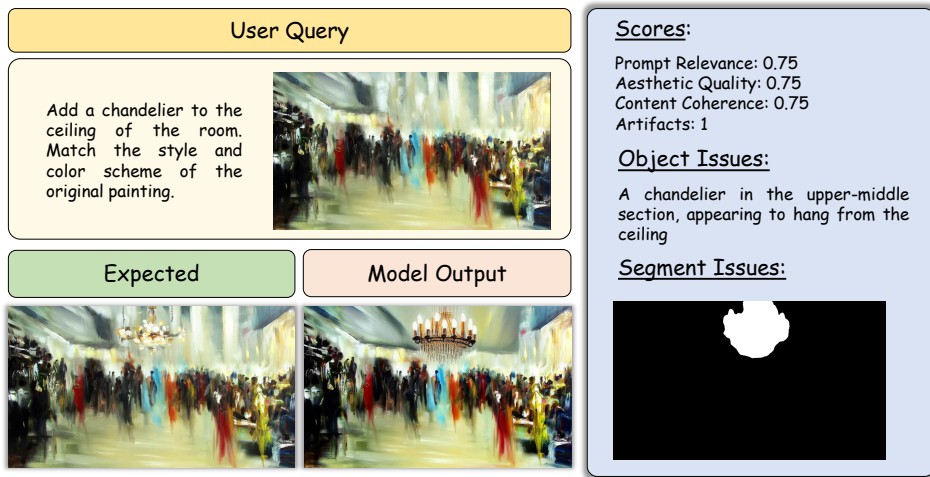

Figure 3: Examples include object-level issues, where expected objects are missing or distorted, and segment-level issues, where SoM partitions highlight specific regions with visual inconsistencies that affect evaluation scores.

## 4  EVALUATION SETUP

**Scoring Criteria.** Our evaluation relies on four criteria that measure Prompt Relevance, Aesthetic Quality, Content Coherence, and Artifact, capturing complementary aspects of image generation and editing quality, similar to prior works (Xu et al., 2023; Ku et al., 2024b). Each criterion is rated on a 5-point Likert scale (1 = poor, 5 = excellent) and later rescaled to the range [0,1]. The definitions of the scoring dimensions are:

- Prompt Relevance: Measures whether the image faithfully reflects the instruction.
- Aesthetic Quality: Evaluates the overall visual appeal and design (e.g., overcrowded or poorly aligned elements, poor color schemes, inconsistent fonts).
- Content Coherence: Assesses logical and semantic consistency (e.g., labels pointing to the wrong region, a chart titled "growth" showing decreasing values, or a figure labeled "Parts of a Flower" depicting tree anatomy).
- Artifacts: Captures visual flaws and technical issues caused by generation errors (e.g., distorted or gibberish text, warped edges, extra limbs, unnatural eyes, or repeated patterns).

Each criterion was evaluated both by human annotators and by automated scorers. Specifically, each image was rated by three annotators independently, while LLM-based scores were obtained using the Gemini-2.5-Flash model following the VIEScore (Ku et al., 2024a) paradigm, which produces ratings aligned with the same four criteria. In addition, CLIPScore (Hessel et al., 2021) and LPIPS (Zhang et al., 2018) were computed as auxiliary automated metrics to assess image quality.

**Explainability via Object and Segment Issues.** While many prior works focus on evaluating image generation quality, few consider the explainability of evaluation scores (Chen et al., 2023; Ku et al., 2024a). To improve interpretability, we define two complementary error taxonomies: **object-level issues** in text and **segment-level issues** in image. In addition to assigning ratings on the four criteria, annotators were asked to identify which objects or regions in the image negatively influenced their scores. For object-level issues, the instruction together with any source or reference images from our dataset were given to Gemini-2.5-Flash, and the model was queried to generate a list of objects expected to appear in the output image. Annotators reviewed this list and marked any objects that were missing, incorrectly rendered, or distorted. For segment-level issues, each generated image was partitioned into regions using the Set-of-Mark (SoM) (Yang et al., 2023), and annotators selected any segments that contained visual flaws or inconsistencies, thereby identifying specific areas of the image responsible for score deductions. Figure 3 illustrates examples of both object-level and segment-level annotations from our dataset.

Table 2: Task coverage and architectural categorization of the evaluated models are shown in chronological order. We classify multimodal large language models as a form of autoregressive model, and group flow matching within the diffusion family, following recent literature (Xiong et al., 2025; Chang et al., 2025). For closed-source models where architectural details are unavailable, we label them as Unknown.

| Model | Architecture | TIG | TIE | SRIG | SRIE | MRIG | MRIE |
|---|---|---|---|---|---|---|---|
| InstructPix2Pix (Brooks et al., 2023b) | Diffusion | ✗ | ✓ | ✗ | ✗ | ✗ | ✗ |
| SDXL (Podell et al., 2023) | Diffusion | ✓ | ✗ | ✗ | ✗ | ✗ | ✗ |
| Infinity (Han et al., 2025) | AR | ✓ | ✗ | ✗ | ✗ | ✗ | ✗ |
| Janus Pro (Chen et al., 2025b) | AR + Diffusion | ✓ | ✗ | ✗ | ✗ | ✗ | ✗ |
| GPT-Image-1 (OpenAI, 2025) | Unknown | ✓ | ✓ | ✓ | ✓ | ✓ | ✓ |
| UNO (Wu et al., 2025c) | AR + Diffusion | ✓ | ✗ | ✓ | ✗ | ✓ | ✗ |
| BAGEL (Deng et al., 2025) | AR | ✓ | ✓ | ✓ | ✓ | ✓ | ✓ |
| Step1X-Edit (Liu et al., 2025) | AR + Diffusion | ✗ | ✓ | ✗ | ✗ | ✗ | ✗ |
| IC-Edit (Zhang et al., 2025) | Diffusion | ✗ | ✓ | ✗ | ✗ | ✗ | ✗ |
| Gemini 2.0 Flash (Wu et al., 2025c) | Unknown | ✓ | ✓ | ✓ | ✓ | ✓ | ✓ |
| OmniGen2 (Wu et al., 2025b) | AR + Diffusion | ✓ | ✓ | ✓ | ✓ | ✓ | ✓ |
| Flux.1-Krea-dev (Lee et al., 2025) | Diffusion | ✓ | ✗ | ✗ | ✗ | ✗ | ✗ |
| Flux.1-Kontext-dev (BlackForestLabs et al., 2025) | Diffusion | ✗ | ✓ | ✓ | ✗ | ✗ | ✗ |
| Qwen-Image (Wu et al., 2025a) | Diffusion | ✓ | ✗ | ✗ | ✗ | ✗ | ✗ |

**Baselines.** We evaluate ImagenWorld using models from three major architectural families: diffusion models, autoregressive models, and hybrids that combine AR with diffusion decoders. Table 2 lists the evaluated models, their architectural families, and their task coverage. This set spans both unified models capable of all six tasks (GPT-Image-1 (Chen et al., 2025a), Gemini 2.0 Flash (Google, 2025), BAGEL (Deng et al., 2025), OmniGen2 (Wu et al., 2025b)) and expert models specialized for subsets of tasks (Wu et al., 2025a; BlackForestLabs et al., 2025; Liu et al., 2025; Han et al., 2025). This setup enables broad comparisons across architectures and between unified and expert approaches.

## 5 RESULTS AND ANALYSIS

We summarize our quantitative findings in Table 3, which reports human evaluation scores and VLM predictions across six tasks and four criteria. The subsections below analyze these results by task, topic, and evaluation metric (see Appendix A.5 for statistical tests).

### 5.1 QUANTITATIVE ANALYSIS

**Task Level.** Across the six benchmark tasks, there is a clear gap between open- and closed-source models as reflected in Figure 4. GPT-Image-1 achieves the strongest overall performance, outperforming Gemini 2.0 Flash by about 0.1–0.2 points on average. This margin is particularly pronounced in editing tasks, where Gemini falls further behind. Despite this average gap, scale alone doesn't determine success: several open-source models (e.g., Flux-Krea-dev, Qwen-Image, Flux-1-Kontext-Dev) outperform Gemini on text-guided generation and editing, showing that larger scale doesn't always yield superior results. However, none of the open-source unified models can catch up with close-source models, as seen in Figure 5. Beyond scale effects, the distribution of scores further highlights differences in task difficulty: models consistently achieve lower performance on editing tasks (TIE, SRIE, MRIE) than on their generation counterparts (TIG, SRIG, MRIG), with an average gap of roughly 0.1 This pattern suggest that while generation has benefited from scaling and improved reasoning integration, localized modification remains a major bottleneck.

**Topic Level.** Performance also varies substantially across topical domains. Figure 4 presents topic-level statistics (see Appendix A.4 for detailed results across the full model set). Among the six defined topics, Artworks (A) and Photorealistic Images (P) stand out as the most successful, with averages approaching 0.78 and the best model (GPT-Image-1) reaching around 0.9 in both categories. This reflects notable progress in rendering high-fidelity content in naturalistic and stylistic domains. In contrast, structured and symbolic topics reveal much larger gaps. Textual Graphics (T) and Computer Graphics (C) both average near 0.68, while Screenshots (S) and Information Graphics (I) remain the most challenging, with averages closer to 0.55. Overall, these findings underscore persistent weaknesses in handling text-heavy and symbolic content, highlighting the need for targeted advances in these domains. We also observe similar trend but more obvious gap in unified models (Figure 5).

Table 3: All evaluated models across six core tasks with bolded maxima for the four human metrics and overall in each task. $\bar{O}_{\text{Human}}$ denotes the human-annotated overall average; $\bar{O}_{\text{VLM}}$ is the VLM-predicted overall. $\alpha$ is Krippendorff's alpha for inter-rater reliability and $\rho_s$ is the Spearman's rank correlation for human-vlm alignment.

| | Human | | | | Overalls | | Agreement | |
| Model | Prompt ↑ Relevance | Aesthetic ↑ Quality | Content ↑ Coherence | Artifacts ↑ | $\bar{O}_{\text{Human}}$ | $\bar{O}_{\text{VLM}}$ | $\alpha$ Kripp. | $\rho_s$ Spear. |
|---|---|---|---|---|---|---|---|---|
| | | | *Text-guided Image Generation (TIG)* | | | | | |
| GPT-Image-1 | **0.90±0.14** | 0.92±0.10 | **0.91±0.13** | **0.92±0.12** | **0.91±0.10** | 0.92±0.13 | 0.52 | 0.39 |
| Qwen-Image | 0.82±0.22 | **0.92±0.11** | 0.88±0.17 | 0.85±0.22 | 0.87±0.15 | 0.86±0.20 | 0.75 | 0.76 |
| Flux.1-Krea-dev | 0.77±0.23 | 0.85±0.16 | 0.84±0.18 | 0.80±0.22 | 0.82±0.17 | 0.82±0.23 | 0.67 | 0.79 |
| Gemini-2.0-Flash | 0.79±0.21 | 0.78±0.21 | 0.83±0.19 | 0.79±0.25 | 0.80±0.19 | 0.82±0.24 | 0.67 | 0.70 |
| UNO | 0.75±0.21 | 0.77±0.19 | 0.83±0.17 | 0.76±0.26 | 0.78±0.18 | 0.72±0.27 | 0.72 | 0.73 |
| BAGEL | 0.72±0.24 | 0.80±0.20 | 0.82±0.21 | 0.73±0.29 | 0.76±0.21 | 0.75±0.26 | 0.74 | 0.78 |
| OmniGen2 | 0.67±0.25 | 0.77±0.23 | 0.78±0.24 | 0.72±0.32 | 0.74±0.23 | 0.75±0.27 | 0.71 | 0.80 |
| Infinity | 0.64±0.30 | 0.74±0.23 | 0.76±0.24 | 0.69±0.31 | 0.70±0.24 | 0.73±0.27 | 0.78 | 0.81 |
| SDXL | 0.55±0.29 | 0.70±0.25 | 0.70±0.27 | 0.61±0.31 | 0.64±0.25 | 0.65±0.30 | 0.75 | 0.79 |
| Janus Pro | 0.62±0.30 | 0.62±0.30 | 0.62±0.30 | 0.60±0.29 | 0.61±0.29 | 0.54±0.32 | 0.77 | 0.74 |
| | | | *Text-guided Image Editing (TIE)* | | | | | |
| GPT-Image-1 | **0.77±0.17** | **0.86±0.15** | **0.86±0.17** | **0.79±0.22** | **0.82±0.15** | 0.80±0.28 | 0.65 | 0.58 |
| Flux.1-Kontext-dev | 0.52±0.31 | 0.76±0.22 | 0.75±0.22 | 0.76±0.26 | 0.70±0.21 | 0.63±0.32 | 0.68 | 0.71 |
| Gemini-2.0-Flash | 0.62±0.29 | 0.68±0.25 | 0.75±0.24 | 0.70±0.28 | 0.69±0.23 | 0.64±0.35 | 0.70 | 0.64 |
| BAGEL | 0.59±0.28 | 0.64±0.26 | 0.73±0.23 | 0.70±0.25 | 0.67±0.22 | 0.59±0.33 | 0.62 | 0.66 |
| OmniGen2 | 0.38±0.31 | 0.63±0.29 | 0.67±0.30 | 0.66±0.31 | 0.58±0.27 | 0.57±0.32 | 0.77 | 0.72 |
| IC-Edit | 0.38±0.28 | 0.54±0.23 | 0.56±0.22 | 0.52±0.24 | 0.50±0.22 | 0.49±0.29 | 0.65 | 0.58 |
| Step1xEdit | 0.34±0.28 | 0.46±0.29 | 0.51±0.29 | 0.51±0.32 | 0.46±0.26 | 0.43±0.36 | 0.73 | 0.71 |
| InstructPix2Pix | 0.26±0.24 | 0.48±0.25 | 0.56±0.26 | 0.51±0.27 | 0.45±0.22 | 0.41±0.30 | 0.64 | 0.66 |
| | | | *Single Reference Image Generation (SRIG)* | | | | | |
| GPT-Image-1 | **0.79±0.20** | **0.86±0.16** | **0.86±0.18** | **0.85±0.18** | **0.84±0.15** | 0.86±0.18 | 0.61 | 0.48 |
| Gemini-2.0-Flash | 0.62±0.26 | 0.65±0.24 | 0.74±0.24 | 0.69±0.28 | 0.67±0.22 | 0.70±0.28 | 0.66 | 0.60 |
| BAGEL | 0.57±0.24 | 0.67±0.25 | 0.69±0.24 | 0.64±0.29 | 0.64±0.22 | 0.65±0.28 | 0.66 | 0.66 |
| UNO | 0.57±0.25 | 0.62±0.23 | 0.65±0.23 | 0.59±0.25 | 0.61±0.21 | 0.60±0.29 | 0.65 | 0.65 |
| OmniGen2 | 0.41±0.26 | 0.61±0.24 | 0.65±0.26 | 0.64±0.28 | 0.58±0.22 | 0.62±0.30 | 0.65 | 0.74 |
| | | | *Single Reference Image Editing (SRIE)* | | | | | |
| GPT-Image-1 | **0.73±0.22** | **0.80±0.20** | **0.86±0.17** | **0.80±0.22** | **0.80±0.16** | 0.76±0.29 | 0.63 | 0.46 |
| Gemini-2.0-Flash | 0.44±0.29 | 0.60±0.24 | 0.69±0.25 | 0.63±0.26 | 0.59±0.21 | 0.54±0.32 | 0.64 | 0.66 |
| BAGEL | 0.35±0.29 | 0.58±0.25 | 0.62±0.26 | 0.65±0.27 | 0.55±0.22 | 0.53±0.32 | 0.56 | 0.60 |
| OmniGen2 | 0.30±0.27 | 0.60±0.27 | 0.62±0.28 | 0.62±0.30 | 0.54±0.24 | 0.54±0.31 | 0.66 | 0.72 |
| | | | *Multiple Reference Image Generation (MRIG)* | | | | | |
| GPT-Image-1 | **0.80±0.17** | **0.86±0.15** | **0.86±0.18** | **0.85±0.17** | **0.84±0.15** | 0.88±0.16 | 0.51 | 0.39 |
| Gemini-2.0-Flash | 0.69±0.25 | 0.72±0.24 | 0.81±0.20 | 0.72±0.29 | 0.73±0.22 | 0.68±0.30 | 0.73 | 0.73 |
| BAGEL | 0.56±0.26 | 0.63±0.26 | 0.66±0.26 | 0.60±0.30 | 0.61±0.24 | 0.59±0.31 | 0.67 | 0.65 |
| OmniGen2 | 0.47±0.26 | 0.66±0.22 | 0.66±0.26 | 0.65±0.28 | 0.61±0.21 | 0.59±0.28 | 0.66 | 0.70 |
| UNO | 0.53±0.22 | 0.61±0.20 | 0.67±0.20 | 0.60±0.22 | 0.60±0.18 | 0.58±0.25 | 0.58 | 0.54 |
| | | | *Multiple Reference Image Editing (MRIE)* | | | | | |
| GPT-Image-1 | **0.72±0.19** | **0.82±0.18** | **0.82±0.20** | **0.80±0.21** | **0.79±0.17** | 0.75±0.27 | 0.66 | 0.55 |
| Gemini-2.0-Flash | 0.49±0.25 | 0.66±0.21 | 0.69±0.22 | 0.64±0.25 | 0.62±0.19 | 0.51±0.31 | 0.62 | 0.56 |
| OmniGen2 | 0.32±0.20 | 0.56±0.25 | 0.55±0.28 | 0.60±0.28 | 0.51±0.22 | 0.45±0.30 | 0.69 | 0.66 |
| BAGEL | 0.28±0.22 | 0.44±0.26 | 0.48±0.29 | 0.51±0.28 | 0.43±0.23 | 0.38±0.28 | 0.67 | 0.64 |

**Evaluation Criteria.** Across the four defined metrics, we observe distinct patterns in model behavior from Figure 4. Prompt Relevance shows the largest variability across tasks, peaking in TIG (0.72) but dropping to 0.46 in editing tasks on average, underscoring the difficulty of aligning edits with instructions. Aesthetic Quality and Content Coherence are more stable, with maximum task-level gaps of 0.17 and 0.16, respectively. Both metrics (Aesthetic Quality/Content Coherence) achieve their highest values in Artworks (0.79/0.79) and Photorealistic Images (0.82/0.82) but decline in symbolic settings such as Screenshots (0.58/0.63) and Information Graphics (0.58/0.59). Artifact suppression appears more uniform at the task level (gap = 0.11), yet topic-level analysis reveals clear asymmetry: non-symbolic content is largely free of distortions, whereas text-heavy categories frequently suffer from unreadable or corrupted elements. Taken together, these results show that instruction following is the primary bottleneck across tasks, while artifact control remains the central challenge in text-intensive domains, particularly Screenshots.

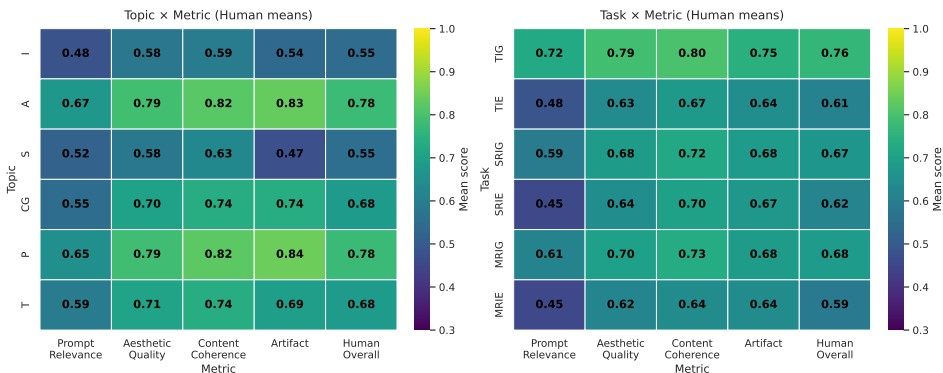

Figure 4: Mean human evaluation scores across metrics by topic (left) and task (right).

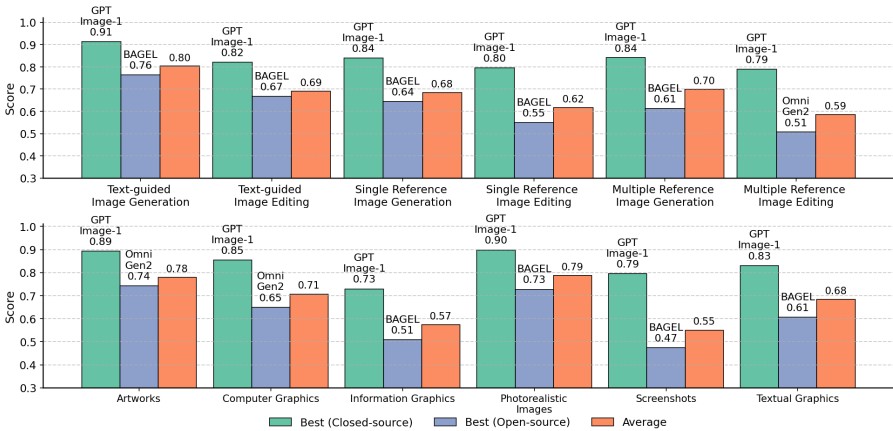

Figure 5: Overall human rating by task and topic for the four unified models that support all six tasks.

## 5.2 QUALITATIVE ANALYSIS

Quantitative metrics alone fail to capture common mistakes that become clear when examining outputs more closely. These mistakes manifest in several recurring ways, as illustrated in Appendix A.3. Across tasks, models often skip parts of complex and multi-step instructions (Figure 8) or produce unreadable and corrupted text (Figure 13), a problem that persists across nearly all text-heavy domains. Numerical inconsistencies are also frequent in symbolic settings, such as pie chart percentages not summing to 100 or receipt totals not matching itemized values (Figure 9). Symbolic and structured domains add further challenges, including errors in understanding depth maps, frequent mismatches between chart legends and plotted data, and incomplete object detection in computer graphics where bounding boxes are poorly aligned with object boundaries (Figure 10). Editing tasks further expose critical weaknesses: models may alter the canvas, misplace objects, neglect shadows and reflections, return the original input unchanged, or even generate a completely new image instead of applying the requested modification (Figure 11). In generation, systems sometimes collapse to near-duplicates of references, ignore one or more references, or produce distorted human figures. These recurring errors clarify why artifact suppression and fine-grained alignment remain key open problems, even for the strongest models.

## 5.3 HUMAN VS. VLM AS JUDGE

To assess the reliability of automatic evaluation, we compare VLM-based judgments with human ratings across all four criteria. Table 4 reports fold-averaged[1] rank correlations (Spearman), Kendall's

---

[1]Fold-averaged results are computed using a leave-one-out procedure: each annotator (or the VLM) is compared against the mean of the other two, and results are averaged across folds.

Table 4: Fold-averaged Spearman correlations, Kendall's accuracy, and Bias (average signed difference between VLM and human ratings) comparing VLM-based judgments with human ratings across all criteria.

| Metric | Human–Human | | VLM–Human | | |
|---|---|---|---|---|---|
| | Spearman $\rho$ | Kendall Acc. | Spearman $\rho$ | Kendall Acc. | Bias |
| Prompt Relevance | $0.67 \pm 0.03$ | $0.77 \pm 0.01$ | $0.70 \pm 0.00$ | $0.79 \pm 0.00$ | $-0.07 \pm 0.01$ |
| Aesthetic Quality | $0.55 \pm 0.01$ | $0.73 \pm 0.00$ | $0.62 \pm 0.01$ | $0.76 \pm 0.00$ | $-0.01 \pm 0.01$ |
| Content Coherence | $0.50 \pm 0.01$ | $0.71 \pm 0.01$ | $0.57 \pm 0.01$ | $0.74 \pm 0.01$ | $-0.05 \pm 0.01$ |
| Artifact | $0.64 \pm 0.01$ | $0.76 \pm 0.01$ | $0.59 \pm 0.01$ | $0.75 \pm 0.00$ | $0.06 \pm 0.00$ |
| Overall | $0.68 \pm 0.02$ | $0.76 \pm 0.01$ | $0.72 \pm 0.00$ | $0.78 \pm 0.00$ | $-0.02 \pm 0.00$ |

accuracy, and VLM bias (the average signed difference between VLM and human ratings). Overall, the VLM exhibits moderate to strong agreement with human raters across evaluation dimensions, aligning well with human–human consistency. Spearman correlations between VLM and human ratings range from 0.57 for content coherence to 0.70 for prompt relevance, closely matching and in some cases exceeding human–human correlations. Kendall's accuracy follows a similar trend, with values between 0.74 and 0.79, indicating that the VLM preserves the relative ranking of samples in line with human judgments. At the metric level, prompt relevance shows the strongest alignment, suggesting that the VLM is particularly effective at assessing semantic faithfulness to the input prompt. In contrast, for artifacts, VLM–human agreement lags behind human–human agreement, and the positive bias indicates that the VLM under-penalizes flaws such as unreadable text, boundary glitches, or misaligned layouts that humans consistently flag. Taken together, the results suggest that VLM-based evaluation provides a reliable proxy for human scoring on high-level criteria such as prompt relevance and aesthetic quality, but human judgment remains crucial for more detailed criteria such as visual defects.

## 6 DISCUSSION

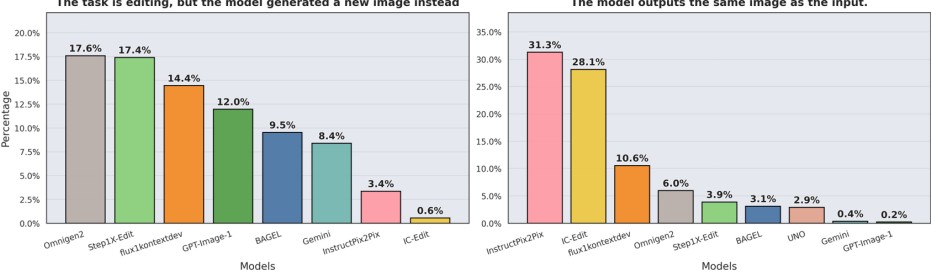

Figure 6: Percentage of cases where the model generating completely new image or simply return input in image editing tasks.

**Targeted data curation boosts domain-specific performance.** While GPT-Image-1 achieves the strongest results across most domains, Qwen-Image demonstrates how deliberate data design can shift the balance in text-heavy scenarios. Unlike many models that struggle with textual content, Qwen-Image consistently outperforms even closed-source systems in artwork, photorealistic images, and especially textual graphics for text-guided generation (Figure 15). This advantage stems from a training pipeline enriched with synthetic text-rich samples and a progressive curriculum that gradually introduced increasingly complex layouts. Together with a balanced domain coverage, these choices directly improved text rendering, making Qwen-Image particularly effective in text-heavy tasks where most models perform poorly.

**Architectural pathways and training objectives influence editing tasks behaviours.** The error rates in Figure 6 suggest that autoregressive–diffusion hybrids such as OmniGen2 and Step1X-Edit are more likely to generate entirely new images when asked to edit (17%), possibly reflecting their reliance on language-driven pathways that can override source conditioning. In contrast, diffusion-

only editors like IC-Edit (0.6%) and InstructPix2Pix (3.4%) rarely disregard the source image, consistent with their architectures being explicitly optimized for localized modifications (though this may come at the cost of limited task coverage). Interestingly, Flux.1 Kontext, despite being diffusion-based, shows a higher rate of generating a new image (14.4%), likely due to its broader training scheme that mixes local edits, global edits, and reference-based composition. This unified in-context design improves versatility but makes the model less constrained to preserve the source image like InstructPix2Pix and IC-Edit.

**Future Work.** The structure of our dataset, which includes human scores, object-level tags, and full model outputs, points to several directions for future research. (1) The collected human scores can be used as preference data to train models that better align with human judgments, for example through preference optimization or ranking-based fine-tuning (Wu et al., 2023; Liang et al., 2024). (2) The object-level tags provide a basis for diagnostic and self-corrective approaches, enabling models to generate object-aware refinements or produce corrective instructions (Chen et al., 2024; Chakrabarty et al., 2023). (3) The combination of fine-grained annotations and human evaluations creates opportunities for developing more interpretable metrics that capture object-level errors, compositional inconsistencies, and prompt–object mismatches more faithfully than existing automatic metrics (Tan et al., 2024; Kirstain et al., 2023b). Taken together, we hope that these directions will support the development of more reliable, controllable, and human-aligned multimodal generation systems.

## 7 CONCLUSION

We present ImagenWorld, a benchmark that unifies six generation and editing tasks across six topical domains, supported by 20K fine-grained human annotations. Unlike prior benchmarks, ImagenWorld introduces an explainable evaluation schema with object-level and segment-level issue labels, enabling interpretable diagnosis of model failures beyond scalar scores. Evaluating 14 models, we find consistent trends: models perform better on generation than on editing, struggle with symbolic and text-heavy domains, and show a persistent gap between closed-source and open-source models. Our structured human annotations capture localized errors and reveal insights that modern VLM-based metrics miss. By combining broad task coverage with explainable labeling, ImagenWorld serves not only as a rigorous benchmark but also as a diagnostic tool, laying the groundwork for more faithful and robust image generation systems.

## ETHICS STATEMENT

The rapid advancement of image generation and editing technologies raises pressing safety and ethical concerns. Traditional photo manipulation tools such as Photoshop already enabled malicious uses, but modern AI-powered systems dramatically lower the technical barrier, making realistic forgeries accessible to a much broader audience. This democratization amplifies risks of fraud and deception in everyday contexts: for example, a user might alter a photo of delivered food to obtain fraudulent refunds, fabricate receipts for economic gain, or generate fake identification documents. More broadly, the same capabilities can be misused for deepfake creation, leading to non-consensual imagery, political disinformation, or privacy violations. Such misuse threatens not only businesses and institutions but also public trust in digital media and the safety of individuals. As generative models continue to improve in fidelity and controllability, it is critical to establish safeguards, encourage responsible deployment, and promote ethical practices. In releasing ImagenWorld, we will provide only sanitized data and annotations, exclude sensitive or personally identifiable content, and make all resources publicly available to support transparent and responsible research.

## REPRODUCIBILITY STATEMENT

All experiments were conducted with 8 NVIDIA A6000 GPUs using the ImagenHub (Ku et al., 2024b) inference library, into which we integrated the latest models when not already included. w For fairness, all model implementations followed the default configurations recommended in their respective papers or official releases, and all open-source models were evaluated under the same seed (42). Approximately $1,000 USD were spent on API access for closed-source models and

VLM-based evaluators. Code, dataset, and sanitized human annotations will be released on publicly accessible platforms (e.g., GitHub, HuggingFace).

## ACKNOWLEDGEMENT

We extend our gratitude to Min-Hung Chen for valuable feedback and insightful comments on this paper. We also sincerely thank Yu-Ying Chiu, Boris Leung, Wyett Zeng, and Dongfu Jiang for their help with data creation and annotation setup.

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

# A APPENDIX

## A.1 USE OF LLM IN WRITING

We used ChatGPT to improve the writing, mainly on grammar fix.

## A.2 HUMAN ANNOTATION DETAILS

During the annotation stage, we recruited 22 expert annotators, primarily graduate students fluent in English. Each annotator evaluated approximately 500–1,000 images, with each annotation taking 30–90 seconds. To reduce fatigue and ensure consistent quality, we restricted annotators to a maximum of 200 samples per week. The entire process was conducted using Label Studio, which provided a user-friendly interface. Potential object-level and segment-level issues were pre-listed using a vision–language model (VLM) and Set-of-Marks (SOM), allowing annotators to simply select them via checkboxes. For cases where the VLM and SOM failed to capture issues, annotators could switch to a manual mode to input text or create bounding boxes. The full annotation process spanned two months, during which all data were collected. Below we present the guidelines for annotators, and Figure 7 illustrates the annotation interface.

---

**Prompt Relevance**

Definition: Whether the image accurately reflects or responds to the prompt.

Ask yourself: Does the image actually follow the instructions or topic in the prompt?

Common issues:

– Prompt: "Explain the water cycle." → Image shows the food chain.

– Prompt: "Slide about diabetes symptoms." → Image shows a kidney structure.

– Prompt: "Compare apples and oranges." → Image compares apples and bananas.

Rating (1–5):

1. Completely unrelated to the prompt.

2. Mostly incorrect; vague connections with many mismatches.

3. Partially relevant; key ideas present but with errors/omissions.

4. Mostly accurate; follows the prompt with minor issues.

5. Fully aligned with the prompt; clear, focused, and complete.

---

**Aesthetic Quality / Visual Appeal**

Definition: Whether the image is visually appealing, clean, and easy to interpret.

Ask yourself: Is this image pleasant to look at, readable, and professionally designed?

Common issues:

– Text too small/hard to read or blending into the background.

– Cluttered or misaligned layout.

– Poor color combinations; inconsistent fonts or spacing.

Rating (1–5):

1. Visually poor; unattractive, hard to read, or confusing.

2. Below average; noticeable design flaws, poor readability.

3. Decent; generally readable with minor layout/design issues.

4. Clean and aesthetically good; few flaws.

5. Polished and visually excellent.

---

**Content Coherence**

Definition: Whether the content is logically consistent and fits together meaningfully.

Ask yourself: Does everything in the image belong together and make sense as a whole?

Common issues:
- Labels pointing to the wrong part (e.g., "liver" pointing to lungs).
- Chart shows decreasing values titled "growth over time."
- Title: "Parts of a Flower" but the figure shows tree anatomy.

Rating (1–5):
1. Internally inconsistent or nonsensical; contradictory parts.
2. Some logic present, but components are confusing/mismatched.
3. Mostly coherent, with noticeable mismatches or awkward parts.
4. Logically sound overall; only minor inconsistencies.
5. Completely coherent and internally consistent.

**Artifacts / Visual Errors**

Definition: Whether the image has visual flaws due to generation errors (e.g., distortions, glitches).

Ask yourself: Are there technical visual issues like melting edges, strange text, or broken parts?

Common issues:
- Gibberish or distorted text (e.g., "#di%et@es" instead of "diabetes").
- Warped borders, duplicated textures, or glitched areas.
- Extra limbs, broken symmetry, over-smoothed or pixelated regions.

Rating (1–5):
1. Severe artifacts that ruin the image.
2. Major flaws that are clearly noticeable.
3. Minor artifacts; image remains usable.
4. Mostly clean; very subtle flaws if any.
5. No visible artifacts.

**Issue Tagging**

- Object-level issues (model output): Select any item that has issues. If none apply, choose "None of the objects have issues."
- Segmentation issues (segmentation map): Select any index with issues. If none apply, choose "None."
- Other issues: If your score is not 5/5 and the issue is not covered above, briefly describe it here.

Important: Selecting "None" may mean either (i) there is no issue, or (ii) your issue is not listed. In the latter case, provide a brief explanation under "Other issues." If your score is not full (5/5), you must either select a corresponding issue or provide a short explanation.

s

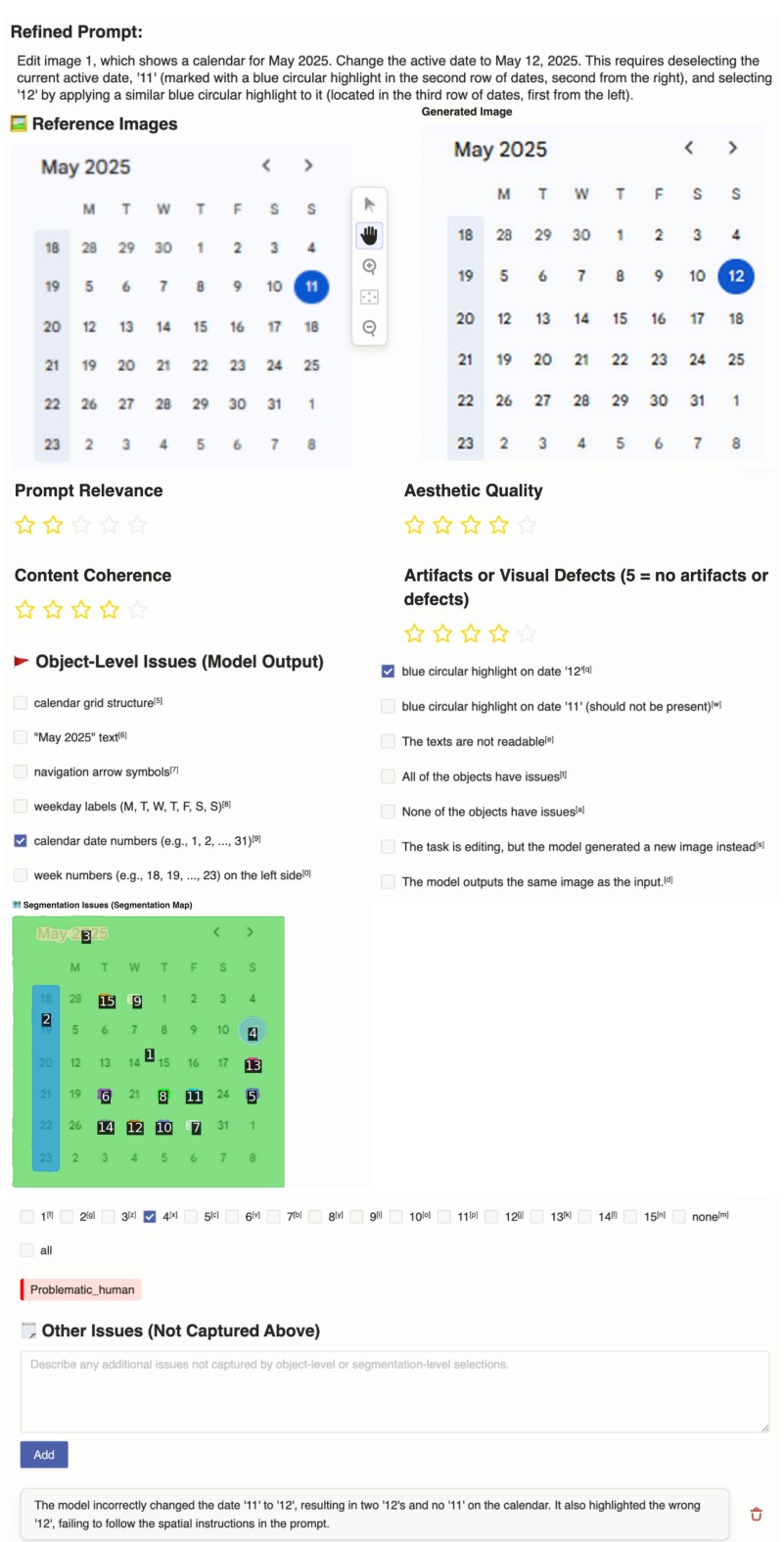

Figure 7: Label Studio interface: annotators read the prompt, inspect the image, rate four criteria (1–5), and flag VLM/SoM-suggested object- and segment-level issues.

## A.3 EXAMPLES OF FAILURES

### A.3.1 MODEL FAILS TO PRECISELY FOLLOW INSTRUCTIONS.

Prompt: Edit image 1. Replace the top-left crate with the yellow warning sign from image 3. Place the pink crewmate (from the center of image 2) and the yellow crewmate (from the bottom right of image 2) standing side-by-side on the central doorway in image 1. Ensure all new elements are integrated with correct perspective, lighting, and scale.

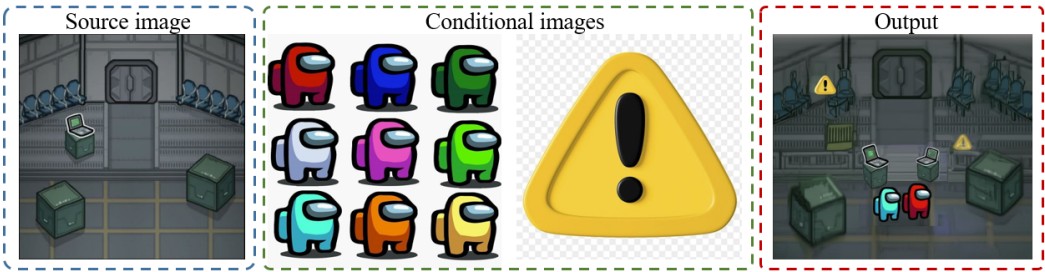

Figure 8: Instruction-following problem: The model placed red and green crewmates instead of pink and yellow, and the yellow sign's position does not match the request.

### A.3.2 NUMERICAL INCONSISTENCIES

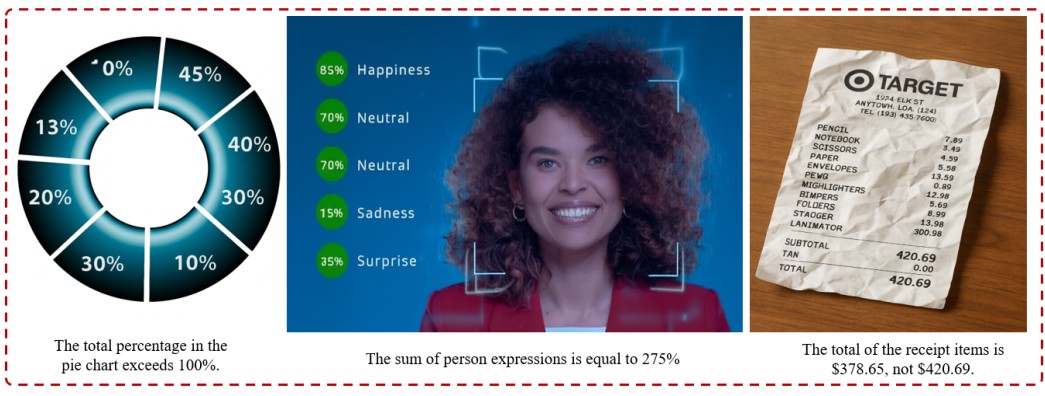

Figure 9: Examples of numerical inconsistencies.

### A.3.3 SEGMENTS AND LABELING ISSUES

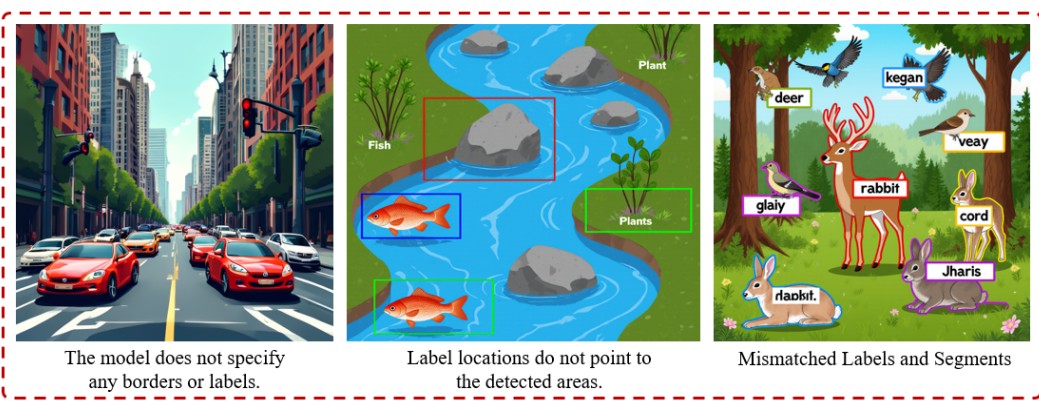

Figure 10: Examples of labeling issues.

### A.3.4   GENERATE NEW IMAGE IN EDITING

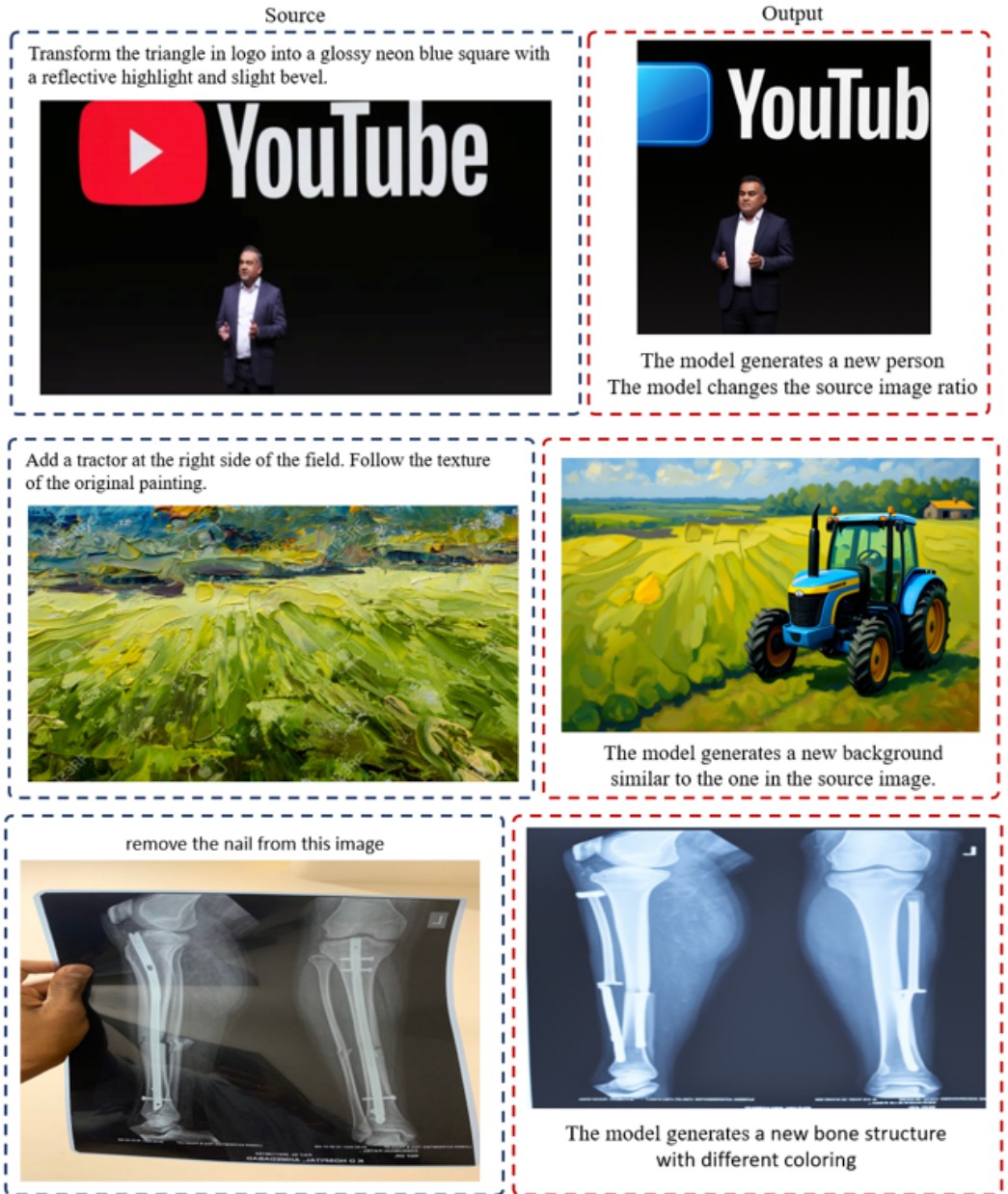

Figure 11: Examples of generating a new image when the task is editing.

### A.3.5 PLOTS AND CHART ERRORS

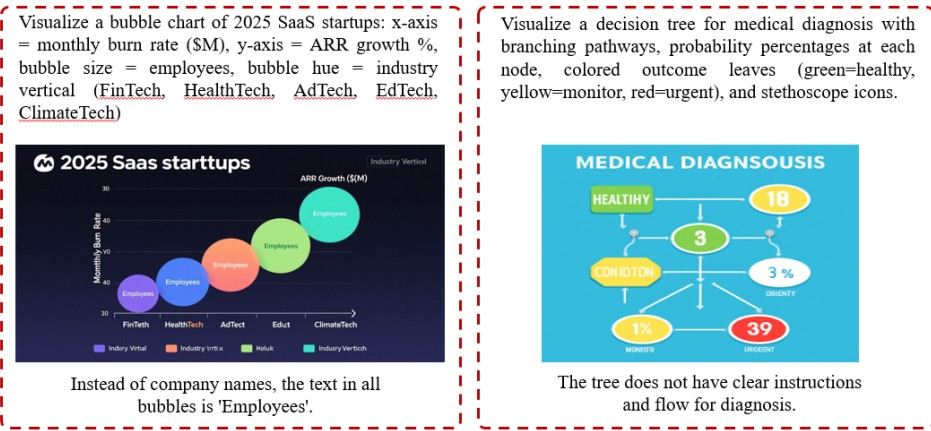

Figure 12: Examples of plots and diagram issues.

### A.3.6 UNREADABLE TEXT

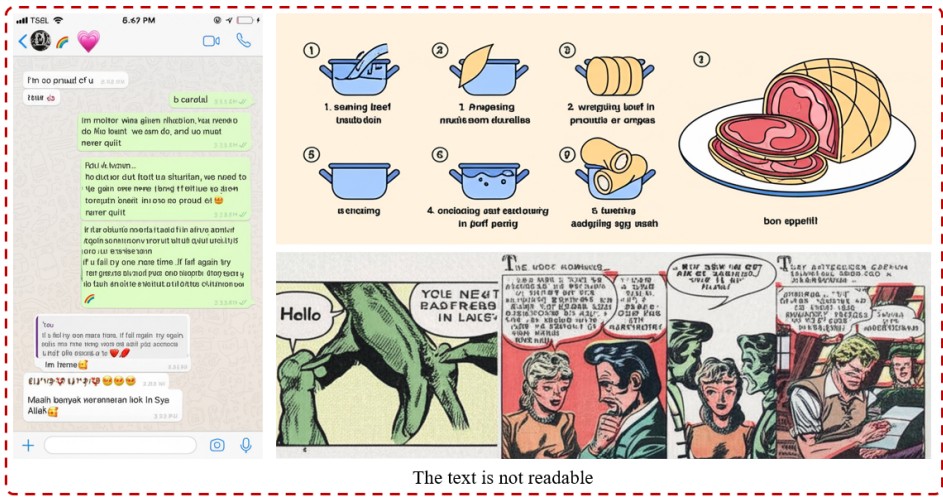

Figure 13: Examples of text issues.

### A.3.7 ERROR MASK STATISTICS

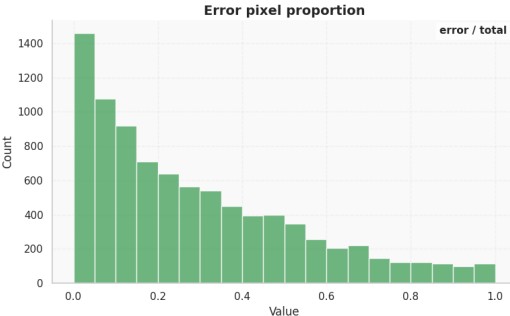

Figure 14: Distribution of error pixel proportions across the dataset. Histograms illustrate the fraction of error pixels relative to total mask pixels, excluding masks with none or all pixels marked as errors.

## A.4 TASK AND TOPIC STATISTICS

| Model | Artworks | Computer Graphics | Information Graphics | Photorealistic Images | Screenshots | Textual Graphics |
|---|---|---|---|---|---|---|
| | | | Text-guided Image Generation | | | |
| GPT-Image-1 | 0.95±0.06 | **0.90±0.09** | **0.81±0.13** | 0.96±0.06 | **0.93±0.09** | 0.92±0.08 |
| Qwen-Image | **0.97±0.03** | 0.81±0.14 | 0.79±0.18 | **0.97±0.07** | 0.73±0.14 | **0.94±0.11** |
| Flux.1-Krea-dev | 0.92±0.06 | 0.75±0.18 | 0.67±0.22 | 0.94±0.08 | 0.73±0.12 | 0.88±0.13 |
| Gemini 2.0 Flash | 0.87±0.09 | 0.73±0.20 | 0.61±0.26 | 0.94±0.09 | 0.75±0.12 | 0.88±0.14 |
| UNO | 0.89±0.08 | 0.73±0.16 | 0.62±0.21 | 0.86±0.13 | 0.64±0.13 | 0.92±0.09 |
| BAGEL | 0.95±0.04 | 0.66±0.19 | 0.66±0.23 | 0.91±0.09 | 0.62±0.17 | 0.79±0.17 |
| Infinity | 0.93±0.08 | 0.69±0.18 | 0.56±0.27 | 0.85±0.15 | 0.44±0.12 | 0.76±0.22 |
| OmniGen2 | 0.86±0.11 | 0.77±0.19 | 0.52±0.29 | 0.89±0.11 | 0.52±0.13 | 0.87±0.13 |
| SDXL | 0.90±0.10 | 0.43±0.27 | 0.55±0.25 | 0.82±0.14 | 0.57±0.18 | 0.57±0.19 |
| Janus Pro | 0.87±0.13 | 0.49±0.27 | 0.42±0.29 | 0.81±0.23 | 0.50±0.22 | 0.60±0.27 |
| | | | Text-guided Image Editing | | | |
| GPT-Image-1 | **0.86±0.09** | **0.90±0.12** | **0.69±0.19** | **0.88±0.09** | **0.73±0.13** | **0.86±0.12** |
| Flux.1-Kontext-dev | 0.73±0.14 | 0.83±0.14 | 0.50±0.23 | 0.76±0.12 | 0.56±0.20 | 0.82±0.17 |
| Gemini 2.0 Flash | 0.72±0.16 | 0.75±0.21 | 0.50±0.23 | 0.83±0.17 | 0.51±0.19 | 0.83±0.18 |
| BAGEL | 0.64±0.20 | 0.86±0.14 | 0.45±0.22 | 0.71±0.16 | 0.61±0.18 | 0.73±0.21 |
| OmniGen2 | 0.74±0.13 | 0.72±0.23 | 0.40±0.21 | 0.81±0.14 | 0.36±0.22 | 0.48±0.24 |
| IC-Edit | 0.59±0.20 | 0.57±0.24 | 0.35±0.18 | 0.55±0.21 | 0.39±0.18 | 0.56±0.21 |
| Step1X-Edit | 0.53±0.26 | 0.39±0.30 | 0.19±0.15 | 0.65±0.19 | 0.47±0.20 | 0.50±0.20 |
| InstructPix2Pix | 0.49±0.22 | 0.53±0.23 | 0.32±0.16 | 0.52±0.25 | 0.48±0.19 | 0.36±0.18 |
| | | | Single Reference Image Generation | | | |
| GPT-Image-1 | **0.95±0.08** | **0.86±0.13** | **0.72±0.18** | **0.87±0.10** | **0.83±0.14** | **0.81±0.15** |
| BAGEL | 0.83±0.17 | 0.76±0.23 | 0.57±0.19 | 0.67±0.16 | 0.52±0.19 | 0.52±0.19 |
| Gemini 2.0 Flash | 0.77±0.19 | 0.73±0.19 | 0.52±0.19 | 0.81±0.19 | 0.56±0.18 | 0.66±0.20 |
| UNO | 0.78±0.20 | 0.65±0.20 | 0.49±0.19 | 0.68±0.20 | 0.56±0.15 | 0.49±0.19 |
| OmniGen2 | 0.78±0.16 | 0.68±0.18 | 0.43±0.19 | 0.66±0.17 | 0.41±0.20 | 0.50±0.16 |
| | | | Single Reference Image Editing | | | |
| GPT-4o | **0.79±0.16** | **0.79±0.18** | **0.80±0.17** | **0.89±0.08** | **0.63±0.15** | **0.86±0.11** |
| Gemini 2.0 Flash | 0.59±0.19 | 0.63±0.22 | 0.39±0.14 | 0.64±0.20 | 0.39±0.14 | 0.63±0.19 |
| BAGEL | 0.54±0.24 | 0.53±0.22 | 0.50±0.26 | 0.68±0.19 | 0.47±0.17 | 0.58±0.19 |
| OmniGen2 | 0.62±0.17 | 0.57±0.23 | 0.27±0.14 | 0.57±0.23 | 0.27±0.14 | 0.47±0.20 |
| | | | Multiple Reference Image Generation | | | |
| GPT-Image-1 | **0.96±0.03** | **0.81±0.12** | **0.66±0.17** | **0.86±0.07** | **0.83±0.12** | **0.92±0.10** |
| Gemini 2.0 Flash | 0.91±0.12 | 0.71±0.20 | 0.53±0.17 | 0.85±0.15 | 0.54±0.15 | 0.86±0.15 |
| OmniGen2 | 0.80±0.14 | 0.62±0.18 | 0.46±0.15 | 0.69±0.17 | 0.40±0.18 | 0.68±0.17 |
| BAGEL | 0.79±0.17 | 0.49±0.22 | 0.47±0.16 | 0.80±0.17 | 0.42±0.20 | 0.71±0.17 |
| UNO | 0.70±0.12 | 0.66±0.14 | 0.41±0.13 | 0.63±0.14 | 0.51±0.15 | 0.71±0.20 |
| | | | Multiple Reference Image Editing | | | |
| GPT-Image-1 | **0.85±0.11** | **0.85±0.10** | **0.69±0.20** | **0.92±0.05** | **0.81±0.12** | **0.62±0.18** |
| Gemini 2.0 Flash | 0.76±0.16 | 0.58±0.17 | 0.62±0.25 | 0.70±0.13 | 0.56±0.16 | 0.50±0.16 |
| OmniGen2 | 0.66±0.13 | 0.53±0.18 | 0.44±0.22 | 0.68±0.19 | 0.32±0.15 | 0.41±0.15 |
| BAGEL | 0.53±0.23 | 0.50±0.20 | 0.41±0.23 | 0.59±0.15 | 0.21±0.11 | 0.32±0.16 |

Table 5: Topic-wise performance across six tasks. Bolded entries are the per-topic best within each task.

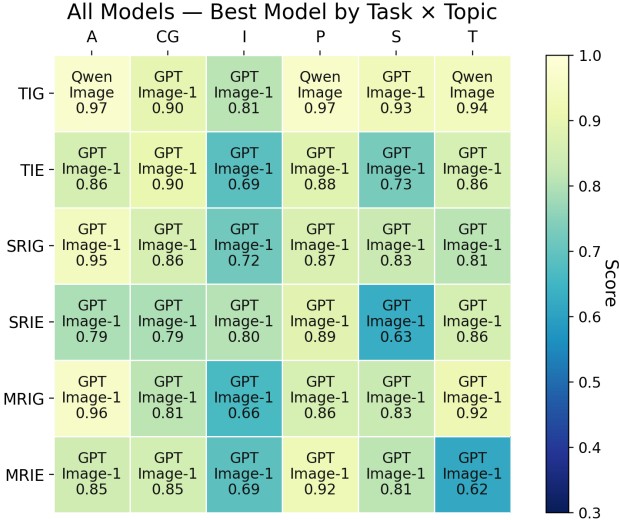

Figure 15: Best-performing model per task×topic across all of our 14 baselines.

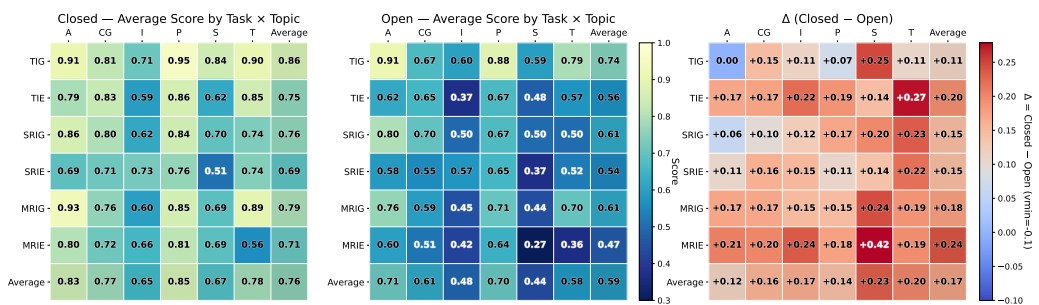

Figure 16: Best-performing model per task×topic for closed-source (left) and open-source (middle) models based on human overall scores; the Δ panel (right) shows Δ = Closed − Open.

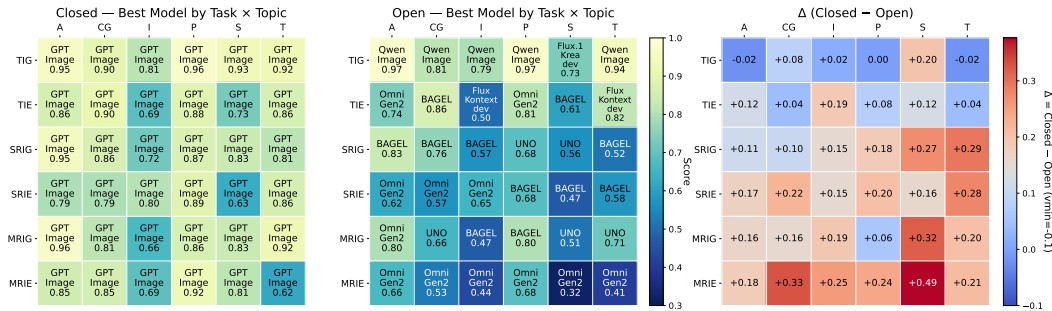

Figure 17: Average human score per task×topic for closed-source (left) and open-source (middle) models; the Δ panel (right) shows Δ = Closed − Open.

## A.5 STATISTICAL SIGNIFICANCE ANALYSIS

We conduct statistical significance tests to quantify the robustness of the performance differences reported in the paper. All analyses operate on human ratings, where each human score denotes the average of the three independent annotators for a given (task_id, model) pair. The tests are performed on a per-item basis. Paired two-sided $t$-tests are used when comparing two models (or model families) evaluated on the same instances, while Welch's $t$-test is used when comparing two disjoint groups of items (e.g., generation vs. editing tasks).

**I. GPT-Image-1 vs. Gemini 2.0 Flash.** We compare GPT-Image-1 and Gemini across all task instances. GPT-Image-1 receives significantly higher ratings overall compared to Gemini 2.0 Flash.

- **Overall**: $n = 1080$, mean diff $= 0.5961$, std $= 0.7706$, $t = 25.42$, $p = 4.3 \times 10^{-112}$.

**II. Closed vs. Open Model Families.** We compare the family-wise means of closed-source models with open-source ones using a paired test across tasks.

- **Overall**: $n = 1080$, mean diff $= 0.6821$, std $= 0.5840$, $t = 38.38$, $p = 6.6 \times 10^{-204}$.

**III. Generation vs. Editing Tasks (Unified Models).** To assess task difficulty, we compare human ratings on generation and editing tasks over our unified models: {Bagel, Gemini 2.0 Flash, GPT-Image-1, OmniGen2}.

- **Overall**: generation ($n = 2158$, mean $= 3.9152$), editing ($n = 2160$, mean $= 3.5244$), $t = 13.71$, $p = 6.6 \times 10^{-42}$.

**IV. Symbolic vs. Non-Symbolic Topics.** We compare human ratings on symbolic and text-heavy domains (Information Graphics, Screenshots) against non-symbolic domains (Artworks, Photorealistic Images).

- **Overall**: non-symbolic ($n = 2159$, mean $= 4.1067$), symbolic ($n = 2157$, mean $= 3.1905$), $t = 34.26$, $p = 7.3 \times 10^{-227}$.
- **Artifacts**: non-symbolic ($n = 2159$, mean $= 4.3522$), symbolic ($n = 2157$, mean $= 3.0163$), $t = 46.23$, $p < 10^{-196}$.

**V. Human vs. VLM-as-Judge Agreement.** We compare Gemini's automatic scores with human-mean ratings for all 6,469 (task_id, model) instances.

- **Prompt Relevance**: mean diff $= -0.2704$, std $= 1.0185$, $t = -21.35$, $p = 8.4 \times 10^{-98}$.
- **Aesthetic Quality**: mean diff $= -0.0529$, std $= 1.0135$, $t = -4.20$, $p = 2.7 \times 10^{-5}$.
- **Content Coherence**: mean diff $= -0.2069$, std $= 1.2741$, $t = -13.06$, $p = 1.7 \times 10^{-38}$.
- **Artifacts**: mean diff $= +0.2220$, std $= 1.1207$, $t = 15.93$, $p = 4.2 \times 10^{-56}$.
- **Overall**: mean diff $= -0.0770$, std $= 0.8516$, $t = -7.28$, $p = 3.9 \times 10^{-13}$.

## A.6 DATASET DETAILS

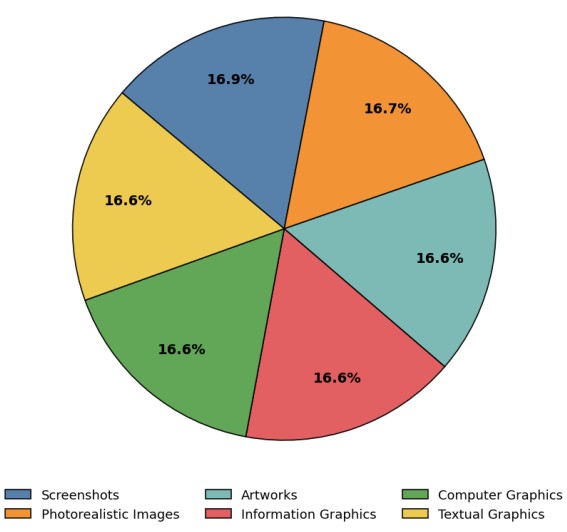

Figure 18: Topic distribution for the entire dataset

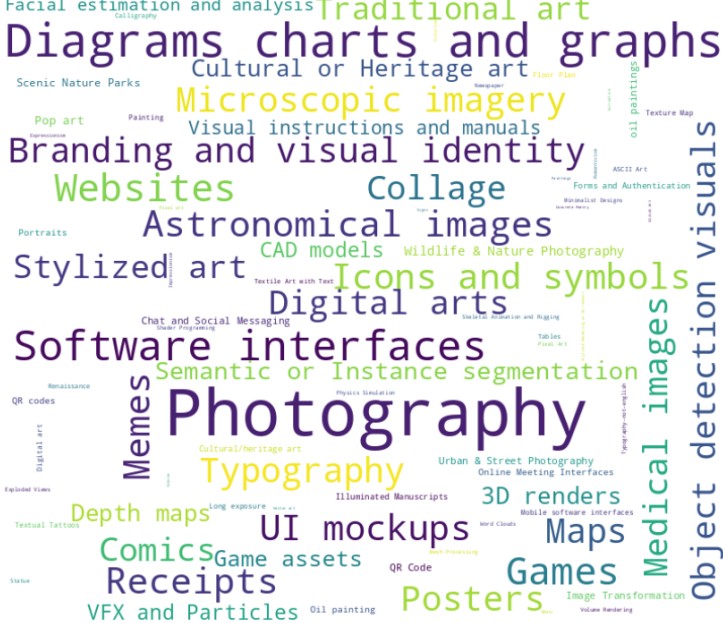

Figure 19: Word cloud of subtopics across the entire dataset, with word size proportional to frequency

| Task | Topic | Subtopics (Count) |
|---|---|---|
| TIG | A | Collage (26), Cultural/heritage art (9), Digital arts (15), Glitch Art (3), Pixel Art (5), Pop art (5), Stylized art (11), Traditional art (11), oil paintings (16) |
| | CG | 3D renders (17), CAD models (13), Depth maps (11), Facial estimation and analysis (3), Game assets (8), Object detection visuals (11), Semantic or Instance segmentation (16), Texture Map (8), Texture maps (1), VFX and Particles (12) |
| | I | Diagrams, charts, and graphs (25), Icons and symbols (26), Maps (17), UI mockups (21), Visual instructions and manuals (11) |
| | P | Astronomical images (18), Long exposure (7), Medical Images (1), Medical images (21), Microscopic imagery (20), Minimalist Designs (5), Photography (28), Retro Futurism (2) |
| | S | Games (19), Receipts (35), Software interfaces (19), Websites (27) |
| | T | ASCII Art (6), Branding and visual identity (7), Calligraphy (6), Comics (13), Concrete Poetry (3), Graffiti (Text-based) (9), Illuminated Manuscripts (9), Memes (11), Posters (12), Textile Art with Text (6), Textual Tattoos (7), Typography (6), Word Clouds (5) |
| SRIG | A | Collage (16), Cultural or Heritage art (14), Digital arts (24), Portraits (1), Stylized art (32), Traditional art (14) |
| | CG | 3D renders (12), CAD models (12), Depth maps (13), Facial estimation and analysis (13), Game Assets (1), Game assets (12), Object detection visuals (13), Semantic or Instance segmentation (13), VFX and Particles (12) |
| | I | Diagrams, charts, and graphs (25), Icons and symbols (21), Maps (14), UI mockups (20), Visual instructions and manuals (20) |
| | P | Astronomical images (28), Medical images (22), Microscopic imagery (23), Photography (27) |
| | S | Games (24), Receipts (18), Software interfaces (28), Websites (30) |
| | T | Branding and visual identity (11), Comics (15), Memes (25), Newspaper (3), Posters (22), Typography (22), Website (2) |
| MRIG | A | Collage (13), Cultural or Heritage art (23), Digital arts (29), Stylized art (19), Traditional art (16) |
| | CG | 3D renders (12), CAD models (11), Depth maps (13), Facial estimation and analysis (10), Game assets (11), Object detection visuals (23), Semantic or Instance segmentation (10), VFX and Particles (10) |
| | I | Diagrams, charts, and graphs (38), Icons and symbols (15), Maps (18), UI mockups (18), Visual instructions and manuals (11) |
| | P | Astronomical images (6), Microscopic imagery (13), Photography (81) |
| | S | Games (24), Receipts (23), Software interfaces (31), Websites (22) |
| | T | Branding and visual identity (24), Comics (15), Memes (16), Posters (23), Typography (18), Typography-not-english (5) |

Table 6: Subtopics across generation tasks (TIG, SRIG, MRIG).

| Task | Topic | Subtopics (Count) |
|---|---|---|
| TIE | A | Animation (2), Blueprint (1), Ceramic art (1), Collage (4), Conceptual art (1), Cropping (1), Cubism (3), Cultural or Heritage art (4), Digital art (8), Digital arts (2), Expressionism (4), Fauvism (2), Futurism (4), Graffiti (1), Graffiti art (1), Hand drawn blueprint (1), Impressionism (4), Medieval art (1), Neoclassicism (2), Oil Painting (1), Painting (8), Paintings (3), Pixel art (4), Pixel arts (2), Pop art (3), Post-Impressionism (2), Renaissance (6), Renaissance arts (1), Romanticism (4), Sketches (2), Statue (5), Surrealism (3), Traditional art (4), Traditional arts (1), Vector art (3), Vector arts (1) |
| | CG | 3D renders (10), CAD models (10), Depth maps (10), Facial estimation and analysis (10), Game assets (10), Mesh Processing (5), Object detection visuals (10), Physics Simulation (5), Semantic or Instance segmentation (10), Shader Programming (5), Skeletal Animation and Rigging (5), VFX and Particles (5), Volume Rendering (5) |
| | I | Diagrams, charts, and graphs (61), Exploded Views (5), Exploded views (2), Floor Plan (8), Icons and symbols (8), Maps (5), Tables (6), UI mockups (5) |
| | P | Astronomical images (21), Medical images (15), Microscopic imagery (20), Photography (45) |
| | S | Game Screenshots (1), Games (26), Mobile software interfaces (7), QR codes (11), Receipts (3), Software interfaces (50), Websites (14) |
| | T | Branding and visual identity (27), Comics (15), Memes (26), Posters (16), Typography (16) |
| SRIE | A | Collage (20), Cultural or Heritage art (19), Digital arts (20), Stylized art (20), Traditional art (21) |
| | CG | 3D renders (11), CAD models (6), Depth maps (13), Facial estimation and analysis (13), Game assets (13), Image Transformation (15), Object detection visuals (18), Semantic or Instance segmentation (11) |
| | I | Diagrams, charts, and graphs (30), Icons and symbols (20), Maps (26), UI mockups (24) |
| | P | Astronomical images (30), Medical images (23), Microscopic imagery (20), Photography (27) |
| | S | Games (25), Receipts (25), Software interfaces (25), Websites (25) |
| | T | Branding and visual identity (22), Comics (22), Memes (20), Posters (16), Typography (20) |
| MRIE | A | Collage (20), Cultural or Heritage art (12), Digital arts (9), Oil painting (12), Pop art (14), Portraits (10), Stylized art (9), Traditional art (14) |
| | CG | 3D Character Integration (1), 3D renders (7), Augmented Reality or Computer Vision Visualization (1), CAD models (1), Computer Animation (1), Computer Animation or Stylized Rendering (1), Depth maps (10), Game assets (9), Geometric Modeling or Stylized Rendering (1), Global Illumination (2), Non-Photorealistic Rendering (1), Object detection visuals (23), Scene Editing (1), Semantic or Instance segmentation (19), Stylized Rendering (1), Stylized Rendering or 3D renders (4), Stylized Rendering or Computer Animation (2), Stylized Rendering or Visual Effects (1), UI Design or Stylized Rendering (1), VFX and Particles (13) |
| | I | Diagrams, charts, and graphs (23), Icons and symbols (22), Maps (25), UI mockups (23), Visual instructions and manuals (8) |
| | P | Astronomical images (15), Medical images (15), Microscopic imagery (10), Scenic Nature Parks (19), Urban & Street Photography (16), Wildlife & Nature Photography (25) |
| | S | Chat and Social Messaging (14), Forms and Authentication (9), Games (14), Online Meeting Interfaces (11), QR Code (14), Receipts (15), Software interfaces (11), Software interfaces - farsi (1), Websites (11) |
| | T | Advertisements (1), Book (1), Branding and visual identity (12), Card (1), Comics (9), Flyers (2), Fonts (1), Magazine (2), Meme (1), Memes (18), Menu (4), Newspaper (2), Newspapers (1), Nutrition Fact (1), Poster (2), Posters (15), Products (1), Signs (5), Subtitle (1), Typography (20) |

Table 7: Subtopics across editing tasks (TIE, SRIE, MRIE).

## A.7 AUTOMATED EVALUATIONS

| Model | LLM-based | | | | | Embedding & perceptual | |
|---|---|---|---|---|---|---|---|
| | Prompt relevance | Aesthetic quality | Content Coherence | Artifact | Overall | CLIP | LPIPS |
| Text-guided Image Generation | | | | | | | |
| GPT-4o | 0.89±0.20 | 0.95±0.12 | 0.91±0.23 | 0.96±0.13 | 0.93±0.13 | 0.25±0.12 | N/A |
| Qwen-Image | 0.79±0.29 | 0.93±0.17 | 0.86±0.28 | 0.88±0.24 | 0.87±0.20 | 0.24±0.12 | N/A |
| Flux.1-Krea-dev | 0.74±0.30 | 0.89±0.20 | 0.82±0.33 | 0.89±0.25 | 0.84±0.23 | 0.24±0.12 | N/A |
| Janus Pro | 0.40±0.31 | 0.52±0.34 | 0.60±0.41 | 0.58±0.38 | 0.52±0.32 | 0.23±0.10 | N/A |
| Gemini 2.0 Flash | 0.79±0.27 | 0.86±0.23 | 0.84±0.31 | 0.83±0.29 | 0.83±0.23 | 0.24±0.11 | N/A |
| BAGEL | 0.65±0.33 | 0.83±0.24 | 0.76±0.37 | 0.81±0.29 | 0.76±0.26 | 0.24±0.12 | N/A |
| UNO | 0.61±0.31 | 0.80±0.25 | 0.79±0.34 | 0.80±0.30 | 0.75±0.25 | 0.24±0.11 | N/A |
| OmniGen2 | 0.62±0.31 | 0.81±0.26 | 0.74±0.38 | 0.81±0.33 | 0.74±0.27 | 0.24±0.12 | N/A |
| Infinity | 0.58±0.33 | 0.79±0.27 | 0.76±0.36 | 0.76±0.33 | 0.72±0.28 | 0.24±0.11 | N/A |
| SDXL | 0.49±0.33 | 0.72±0.31 | 0.71±0.39 | 0.72±0.35 | 0.66±0.30 | 0.24±0.11 | N/A |
| Janus Pro | 0.40±0.31 | 0.52±0.34 | 0.60±0.41 | 0.58±0.38 | 0.52±0.32 | 0.23±0.10 | N/A |
| Text-guided Image Editing | | | | | | | |
| GPT-4o | 0.77±0.31 | 0.81±0.30 | 0.79±0.35 | 0.84±0.32 | 0.80±0.28 | 0.25±0.10 | 0.54±0.15 |
| Flux.1-Kontext-dev | 0.52±0.38 | 0.66±0.34 | 0.67±0.41 | 0.76±0.36 | 0.65±0.31 | 0.25±0.10 | 0.52±0.18 |
| BAGEL | 0.56±0.37 | 0.61±0.35 | 0.68±0.39 | 0.70±0.37 | 0.64±0.32 | 0.25±0.09 | 0.33±0.23 |
| Gemini 2.0 Flash | 0.57±0.37 | 0.61±0.36 | 0.63±0.42 | 0.68±0.40 | 0.62±0.34 | 0.25±0.09 | 0.39±0.23 |
| OmniGen2 | 0.34±0.35 | 0.56±0.35 | 0.61±0.43 | 0.72±0.39 | 0.56±0.32 | 0.25±0.09 | 0.42±0.25 |
| IC-Edit | 0.25±0.34 | 0.50±0.35 | 0.63±0.41 | 0.66±0.40 | 0.51±0.30 | 0.25±0.10 | 0.23±0.19 |
| InstructPix2Pix | 0.16±0.27 | 0.45±0.35 | 0.58±0.43 | 0.61±0.43 | 0.45±0.30 | 0.25±0.09 | 0.27±0.19 |
| Step1X-Edit | 0.35±0.34 | 0.44±0.37 | 0.46±0.43 | 0.53±0.42 | 0.44±0.34 | 0.25±0.09 | 0.36±0.27 |
| Single Reference Image Generation | | | | | | | |
| GPT-Image-1 | 0.77±0.28 | 0.91±0.16 | 0.88±0.24 | 0.93±0.20 | 0.88±0.17 | 0.15±0.09 | 0.73±0.08 |
| Gemini 2.0 Flash | 0.61±0.32 | 0.73±0.35 | 0.73±0.38 | 0.77±0.35 | 0.69±0.28 | 0.17±0.10 | 0.59±0.16 |
| BAGEL | 0.51±0.30 | 0.68±0.32 | 0.65±0.42 | 0.73±0.37 | 0.64±0.30 | 0.17±0.10 | 0.68±0.17 |
| UNO | 0.40±0.30 | 0.65±0.27 | 0.66±0.40 | 0.73±0.37 | 0.61±0.29 | 0.16±0.10 | 0.66±0.14 |
| OmniGen2 | 0.41±0.33 | 0.63±0.35 | 0.64±0.42 | 0.74±0.37 | 0.61±0.31 | 0.17±0.11 | 0.53±0.24 |
| Single Reference Image Editing | | | | | | | |
| GPT-Image-1 | 0.75±0.30 | 0.80±0.29 | 0.78±0.35 | 0.84±0.32 | 0.79±0.27 | 0.24±0.10 | 0.66±0.12 |
| Gemini 2.0 Flash | 0.40±0.35 | 0.55±0.35 | 0.59±0.42 | 0.67±0.39 | 0.55±0.32 | 0.24±0.10 | 0.59±0.16 |
| OmniGen2 | 0.30±0.30 | 0.59±0.34 | 0.58±0.43 | 0.70±0.40 | 0.54±0.30 | 0.24±0.11 | 0.69±0.11 |
| BAGEL | 0.31±0.33 | 0.56±0.35 | 0.59±0.42 | 0.67±0.38 | 0.53±0.30 | 0.24±0.10 | 0.57±0.16 |
| Multiple Reference Image Generation | | | | | | | |
| GPT-Image-1 | 0.69±0.31 | 0.75±0.30 | 0.73±0.37 | 0.80±0.33 | 0.74±0.29 | 0.16±0.10 | 0.72±0.07 |
| UNO | 0.35±0.24 | 0.64±0.27 | 0.62±0.39 | 0.73±0.36 | 0.59±0.26 | 0.13±0.09 | 0.73±0.09 |
| Gemini 2.0 Flash | 0.35±0.30 | 0.49±0.35 | 0.50±0.42 | 0.59±0.41 | 0.48±0.32 | 0.17±0.10 | 0.69±0.09 |
| OmniGen2 | 0.26±0.28 | 0.51±0.34 | 0.45±0.42 | 0.63±0.40 | 0.46±0.30 | 0.16±0.11 | 0.72±0.09 |
| BAGEL | 0.23±0.28 | 0.39±0.34 | 0.36±0.40 | 0.49±0.40 | 0.37±0.29 | 0.14±0.09 | 0.70±0.09 |
| Multiple Reference Image Editing | | | | | | | |
| GPT-Image-1 | 0.69±0.31 | 0.75±0.30 | 0.73±0.37 | 0.80±0.33 | 0.74±0.29 | 0.16±0.10 | 0.72±0.07 |
| Gemini 2.0 Flash | 0.35±0.30 | 0.49±0.35 | 0.50±0.42 | 0.59±0.41 | 0.48±0.32 | 0.17±0.10 | 0.69±0.09 |
| OmniGen2 | 0.26±0.26 | 0.51±0.34 | 0.45±0.42 | 0.63±0.40 | 0.46±0.30 | 0.16±0.11 | 0.73±0.07 |
| BAGEL | 0.23±0.24 | 0.39±0.34 | 0.36±0.40 | 0.49±0.40 | 0.37±0.29 | 0.16±0.10 | 0.70±0.09 |

Table 8: Automatic evaluations on all entries in our dataset

| Model | LLM-based | | | | | Embedding & perceptual | |
|---|---|---|---|---|---|---|---|
| | Prompt relevance | Aesthetic quality | Content Coherence | Artifact | Overall | CLIP | LPIPS |
| Text-guided Image Generation | | | | | | | |
| GPT-Image-1 | 0.88±0.22 | 0.95±0.13 | 0.91±0.23 | 0.96±0.13 | 0.92±0.13 | 0.24±0.12 | N/A |
| Qwen-Image | 0.77±0.31 | 0.93±0.17 | 0.87±0.28 | 0.89±0.24 | 0.86±0.20 | 0.23±0.12 | N/A |
| Flux.1-Krea-dev | 0.72±0.31 | 0.88±0.21 | 0.80±0.35 | 0.89±0.26 | 0.82±0.23 | 0.23±0.12 | N/A |
| Gemini 2.0 Flash | 0.78±0.28 | 0.85±0.23 | 0.82±0.31 | 0.83±0.29 | 0.82±0.24 | 0.24±0.12 | N/A |
| BAGEL | 0.62±0.34 | 0.83±0.23 | 0.74±0.37 | 0.82±0.30 | 0.75±0.26 | 0.23±0.12 | N/A |
| OmniGen2 | 0.62±0.33 | 0.81±0.25 | 0.73±0.37 | 0.82±0.31 | 0.75±0.27 | 0.22±0.12 | N/A |
| UNO | 0.57±0.32 | 0.79±0.26 | 0.75±0.37 | 0.78±0.31 | 0.72±0.27 | 0.23±0.12 | N/A |
| Infinity | 0.58±0.32 | 0.80±0.26 | 0.76±0.36 | 0.75±0.34 | 0.73±0.27 | 0.22±0.11 | N/A |
| SDXL | 0.48±0.33 | 0.72±0.32 | 0.71±0.39 | 0.71±0.36 | 0.65±0.30 | 0.23±0.11 | N/A |
| Janus Pro | 0.41±0.30 | 0.55±0.33 | 0.62±0.41 | 0.60±0.37 | 0.54±0.32 | 0.22±0.10 | N/A |
| Text-guided Image Editing | | | | | | | |
| GPT-Image-1 | 0.77±0.31 | 0.81±0.30 | 0.79±0.35 | 0.85±0.31 | 0.80±0.28 | 0.24±0.10 | 0.55±0.15 |
| Gemini 2.0 Flash | 0.59±0.37 | 0.62±0.37 | 0.66±0.41 | 0.68±0.40 | 0.64±0.35 | 0.24±0.10 | 0.42±0.23 |
| Flux.1-Kontext-dev | 0.48±0.36 | 0.66±0.35 | 0.64±0.42 | 0.75±0.37 | 0.63±0.32 | 0.24±0.10 | 0.52±0.19 |
| BAGEL | 0.51±0.37 | 0.57±0.37 | 0.65±0.41 | 0.64±0.39 | 0.59±0.33 | 0.24±0.10 | 0.35±0.25 |
| OmniGen2 | 0.34±0.36 | 0.58±0.35 | 0.63±0.43 | 0.74±0.38 | 0.57±0.32 | 0.25±0.10 | 0.41±0.26 |
| IC-Edit | 0.23±0.32 | 0.48±0.35 | 0.60±0.40 | 0.64±0.41 | 0.49±0.29 | 0.24±0.09 | 0.23±0.18 |
| Step1X-Edit | 0.34±0.35 | 0.43±0.38 | 0.45±0.45 | 0.52±0.43 | 0.43±0.36 | 0.24±0.09 | 0.39±0.30 |
| InstructPix2Pix | 0.14±0.25 | 0.41±0.35 | 0.52±0.43 | 0.58±0.44 | 0.41±0.30 | 0.24±0.09 | 0.26±0.19 |
| Single Reference Image Generation | | | | | | | |
| GPT-Image-1 | 0.73±0.28 | 0.91±0.16 | 0.87±0.24 | 0.92±0.22 | 0.86±0.18 | 0.16±0.10 | 0.73±0.18 |
| Gemini 2.0 Flash | 0.58±0.33 | 0.72±0.35 | 0.74±0.38 | 0.76±0.35 | 0.70±0.28 | 0.16±0.10 | 0.59±0.16 |
| BAGEL | 0.49±0.30 | 0.68±0.32 | 0.65±0.42 | 0.73±0.37 | 0.65±0.28 | 0.16±0.10 | 0.69±0.17 |
| OmniGen2 | 0.41±0.33 | 0.63±0.35 | 0.64±0.42 | 0.74±0.37 | 0.62±0.30 | 0.17±0.11 | 0.55±0.24 |
| UNO | 0.38±0.29 | 0.64±0.32 | 0.65±0.40 | 0.73±0.37 | 0.60±0.29 | 0.15±0.10 | 0.68±0.14 |
| Single Reference Image Editing | | | | | | | |
| GPT-4o | 0.67±0.30 | 0.77±0.29 | 0.78±0.35 | 0.84±0.32 | 0.76±0.29 | 0.24±0.10 | 0.65±0.13 |
| OmniGen2 | 0.29±0.31 | 0.58±0.34 | 0.58±0.43 | 0.70±0.40 | 0.54±0.31 | 0.24±0.10 | 0.69±0.11 |
| BAGEL | 0.38±0.35 | 0.54±0.36 | 0.60±0.42 | 0.65±0.38 | 0.53±0.32 | 0.24±0.10 | 0.56±0.17 |
| Gemini 2.0 Flash | 0.42±0.36 | 0.52±0.37 | 0.59±0.42 | 0.68±0.40 | 0.51±0.32 | 0.24±0.10 | 0.59±0.16 |
| Multiple Reference Image Generation | | | | | | | |
| GPT-Image-1 | 0.79±0.25 | 0.91±0.17 | 0.90±0.25 | 0.93±0.21 | 0.88±0.16 | 0.14±0.09 | 0.73±0.08 |
| Gemini 2.0 Flash | 0.59±0.31 | 0.70±0.30 | 0.70±0.39 | 0.74±0.37 | 0.68±0.30 | 0.14±0.09 | 0.74±0.09 |
| OmniGen2 | 0.40±0.28 | 0.64±0.28 | 0.58±0.40 | 0.74±0.38 | 0.59±0.28 | 0.13±0.10 | 0.72±0.09 |
| BAGEL | 0.47±0.26 | 0.61±0.33 | 0.61±0.41 | 0.68±0.39 | 0.59±0.31 | 0.14±0.09 | 0.74±0.08 |
| UNO | 0.35±0.25 | 0.65±0.26 | 0.62±0.39 | 0.72±0.36 | 0.58±0.25 | 0.13±0.09 | 0.74±0.08 |
| Multiple Reference Image Editing | | | | | | | |
| GPT-Image-1 | 0.67±0.31 | 0.77±0.30 | 0.73±0.36 | 0.82±0.32 | 0.75±0.28 | 0.16±0.10 | 0.72±0.08 |
| Gemini 2.0 Flash | 0.36±0.30 | 0.51±0.35 | 0.53±0.42 | 0.62±0.41 | 0.51±0.31 | 0.17±0.10 | 0.69±0.09 |
| OmniGen2 | 0.23±0.23 | 0.50±0.34 | 0.42±0.42 | 0.64±0.41 | 0.45±0.30 | 0.15±0.11 | 0.73±0.07 |
| BAGEL | 0.23±0.25 | 0.40±0.33 | 0.37±0.39 | 0.52±0.40 | 0.38±0.28 | 0.16±0.10 | 0.70±0.09 |

Table 9: Automatic evaluations on our human-annotated subset

## A.8 PROMPT TEMPLATES

---

**Prompt Template for Image Generation & Editing**

You are an expert visual generation assistant.
Task: {TASK_NAME}
Task Definition: {TASK_DEFINITION}
Visual Domain: {TOPIC_NAME}
User Objective: {USER_PROMPT}
Attached Images: {IMG_LIST} (None for Text-guided image generation)
Please generate an image that fulfills the user's objective, adheres to the task definition, and fits within the specified visual domain.

TASK_NAME:
- Text-guided Image Generation
- Text-guided Image Editing
- Single Reference-guided Image Generation
- Single Reference-guided Image Editing
- Multiple References-guided Image Generation
- Multiple References-guided Image Editing

TASK_DEFINITION:
- Text-guided Image Generation: Generate a completely new image based only on a descriptive text prompt. No source or reference images are provided.
- Text-guided Image Editing: Edit an existing image using a descriptive text prompt. Decide what to modify in the image based on the prompt. No mask or marked region is given.
- Single Reference-guided Image Generation: Create a new image by combining visual cues from one reference image with instructions from a descriptive text prompt.
- Single Reference-guided Image Editing: Edit an existing image using both a reference image and a text prompt. Use the reference image to guide the style or content of the edits.
- Multiple References-guided Image Generation: Generate a new image using several reference images along with a text prompt. The new image should reflect visual elements from the references and follow the prompt's description.
- Multiple References-guided Image Editing: Modify an existing image using multiple reference images and a descriptive text prompt. The edits should be guided by both the style or content of the references and the instructions in the prompt.

TOPIC_NAME:
- Information Graphics
- Artworks
- Screenshots
- Computer Graphics
- Photorealistic Images
- Textual Graphics

---

---

**Prompt Template for Image Evaluation**

**You are an expert AI image evaluator.** Your task is to rate a generated image based on a provided text prompt and any reference images.

Use the following guidelines for your assessment. Provide a rating from 1 to 5 for each criterion. Do **NOT** include any additional text or explanations in your response. The response **MUST** be a single JSON object.

**Quality Assessment**

- **Prompt Relevance**
  Definition: Whether the image accurately reflects or responds to the prompt.
  *Rating Guide (1–5):*
  1 – Completely unrelated to the prompt.
  2 – Mostly incorrect; some vague connections but many mismatches.
  3 – Partially relevant; key ideas are present but with errors or omissions.
  4 – Mostly accurate; follows the prompt well with minor issues.
  5 – Fully aligned with the prompt; clear, focused, and complete.

- **Aesthetic Quality / Visual Appeal**
  Definition: Whether the image is visually appealing, clean, and easy to interpret.
  *Rating Guide (1–5):*
  1 – Visually poor; unattractive, hard to read or confusing.
  2 – Below average; noticeable design flaws, poor readability.
  3 – Decent; generally readable but has minor layout/design issues.
  4 – Clean and aesthetically good; professional feel with few flaws.
  5 – Beautiful, polished, and visually excellent.

- **Content Coherence**
  Definition: Whether the content in the image is logically consistent and fits together meaningfully.
  *Rating Guide (1–5):*
  1 – Internally inconsistent or nonsensical; parts contradict each other.
  2 – Some logic, but confusing or mismatched components.
  3 – Mostly coherent, though there are noticeable mismatches or awkward parts.
  4 – Logically sound overall, with only minor inconsistencies.
  5 – Completely coherent and internally consistent.

- **Artifacts / Visual Errors**
  Definition: Whether the image has visual flaws due to generation errors (e.g., distortions, glitches).
  *Rating Guide (1–5):*
  1 – Severe artifacts that ruin the image.
  2 – Major flaws that are clearly noticeable.
  3 – Some minor artifacts, but the image remains usable.
  4 – Mostly clean; only very subtle flaws if any.
  5 – Perfectly clean; no visible artifacts at all.

**Expected Output (single JSON object):**

```
{
  "prompt_relevance": <rating>,
  "aesthetic_quality": <rating>,
  "content_coherence": <rating>,
  "artifacts": <rating>
}
```

