# OpenReview forum: "ImagenWorld: Stress-Testing Image Generation Models with Explainable Human Evaluation on Open-ended Real-World Tasks"
_ICLR.cc/2026/Conference — ICLR 2026 Poster_

### Official Review · Reviewer_7NZP · 2025-10-30

**Soundness:** 3
**Presentation:** 4
**Contribution:** 3
**Rating:** 8
**Confidence:** 5

**Summary:**

This paper introduces ImagenWorld, a new large-scale benchmark designed to "stress-test" modern image generation and editing models.  ImagenWorld is built on a comprehensive foundation comprising:
- Six Core Tasks: It covers a wide range of functionalities, including text-guided image generation, single and multiple reference-guided generation, and the corresponding editing variations of these tasks.
- Six Topical Domains: The benchmark includes a diverse set of real-world use cases: artworks, photorealistic images, information graphics, textual graphics, computer graphics, and screenshots.
- Explainable Human Evaluation: At its core, ImagenWorld relies on a detailed human evaluation schema. Instead of just providing a single score, annotators identify and tag specific, localized errors at both the object level (e.g., a "missing chandelier") and the segment level (e.g., visual inconsistencies in a specific region of the image). This is supported by 3.6K condition sets and 20K fine-grained human annotations.

The authors conduct a large-scale study evaluating 14 different models, including closed-source systems like GPT-Image-1 and Gemini 2.0 Flash, as well as various open-source models. The key findings from this evaluation are:
- Models generally struggle more with editing tasks than generation, often failing by either completely regenerating the image or ignoring the edit instruction entirely.
- All models excel in artistic and photorealistic domains but perform poorly on text-heavy and symbolic domains like screenshots and information graphics. An exception is Qwen-Image, whose performance suggests that targeted data curation can mitigate this weakness.
- While closed-source models lead in overall performance, open-source models are competitive in standard text-to-image generation, indicating that progress in more complex editing and multimodal tasks requires more than just scaling.
- Modern Vision-Language Models (VLMs) used as automated judges achieve high correlation with human rankings but fail to provide the fine-grained, explainable error analysis that human annotators can.

**Strengths:**

1. The paper presents a valuable benchmark and evaluation protocol for image generation models. As the capabilities of the models are improving and we are moving to models that can perform several different tasks from text-to-image generation to multi-image reference image editing unifying evaluation is important.
2. The authors systematically evaluate several closed- and open-source models and draw conclusions on the current weaknesses of image generation models. They make sure to include different model families and measure performance for different tasks and with respect to different generation aspects (prompt relevance, content coherence, etc)
3. The authors assess the reliability of automatic judges by measuring correlation with human ratings and find that there is moderate to strong correlation with coarse-grained ratings per generation aspect.

**Weaknesses:**

1. One important contribution of the paper is the collection of fine-grained feedback on the image generation errors by human raters by annotating the object and segment errors in the images. These annotations are not being sufficiently utilized in the paper for either analysis or automatic judge improvement. The authors mention is Section 5.3 that "Taken together, the results suggest that VLM-based evaluation provides a reliable proxy for human annotation, particularly for assessing semantic alignment and visual appeal, while human judgment remains essential for identifying fine-grained details such as visual defects." without providing any analysis on the fine-grained collected data (they only present results and correlation on the coarse-grained ratings). Moreover, it would be great to show whether a subset of these data can be used as a few shot prompt or for fine-tuning in order to improve the capabilities of VLMs on evaluating fine-grained errors in image generation.

**Questions:**

1. Regarding to the weakness point mentioned above, how do you conclude that VLMs cannot provide more fine-grained details? Can this be improved by simply giving few-shot examples or fine-tuning the evaluator to create fine-grained criteria lists?

---

> ### Author Response · Authors · 2025-11-24
> **Author Response**
>
> We sincerely thank the reviewer for their positive feedback and their insightful comments on our paper. Below, we provide responses to each concern or question.
> ***
> ***Weaknesses:***
> > One important contribution of the paper is the collection of fine-grained feedback on the image generation errors by human raters by annotating the object and segment errors in the images. These annotations are not being sufficiently utilized in the paper for either analysis or automatic judge improvement. The authors mention is Section 5.3 that "Taken together, the results suggest that VLM-based evaluation provides a reliable proxy for human annotation, particularly for assessing semantic alignment and visual appeal, while human judgment remains essential for identifying fine-grained details such as visual defects." without providing any analysis on the fine-grained collected data (they only present results and correlation on the coarse-grained ratings). Moreover, it would be great to show whether a subset of these data can be used as a few shot prompt or for fine-tuning in order to improve the capabilities of VLMs on evaluating fine-grained errors in image generation.
>
> The fine-grained annotations indeed introduce additional complexity, and a deeper analysis of them is an important direction for future work. Our goal in collecting these annotations was to capture how humans identify generation errors and to highlight where current models fail, as illustrated in the Qualitative Analysis and Discussion section, with further detailed examples and error-mask distribution provided in Appendix A.3. In Section 5.3, our discussion focuses on the primary metrics of the benchmark. Semantic alignment and visual appeal correspond to prompt relevance and aesthetic quality, which provide more holistic evaluations, while visual defects reflect artifact-level issues that are inherently more fine-grained. We have revised the sentence to clarify this distinction and avoid confusion.
>
> ***
> ***Questions:***
> > Regarding to the weakness point mentioned above, how do you conclude that VLMs cannot provide more fine-grained details? Can this be improved by simply giving few-shot examples or fine-tuning the evaluator to create fine-grained criteria lists?
>
> We define artifact detection as capturing fine-grained issues (e.g., extra limbs, distorted eyes, glitches), which is more detailed than prompt relevance or aesthetic quality. Artifact detection is the only criterion for which both Spearman correlation and Kendall accuracy were lower for the automated judge. Moreover, in a subset of 214 extreme examples restricted to broader issue types where all three annotators fully agreed (e.g., unreadable text, generating a new image during editing, ignored references), the model detected only 75.7 percent of cases. While adding a few illustrative examples might seem helpful, prior work shows that few-shot prompting does not meaningfully improve visual evaluation[1,2]. However, we believe fine-tuning can help models provide detailed and explainable traces[3] and we would further explore it in our feature work.
> ***
> ***References:***
>
> [1] Ku, Max, et al. "Viescore: Towards explainable metrics for conditional image synthesis evaluation." Proceedings of the 62nd Annual Meeting of the Association for Computational Linguistics (Volume 1: Long Papers). 2024.
>
> [2] Chen, Shuo, et al. "True multimodal in-context learning needs attention to the visual context."  Second Conference on Language Modeling. 2025.
>
> [3] Guo, Yuxiang, et al. "ImageDoctor: Diagnosing Text-to-Image Generation via Grounded Image Reasoning." arXiv preprint arXiv:2510.01010 (2025).

---

### Official Review · Reviewer_QiZo · 2025-10-30

**Soundness:** 3
**Presentation:** 4
**Contribution:** 3
**Rating:** 6
**Confidence:** 4

**Summary:**

The paper introduces Imagen World, a large-scale benchmark designed to stress-test generative image models across a broad spectrum of 3,600 condition sets, covering six core tasks (generation and editing with varying reference guidance) and six topical domains. To address the limitations of opaque scalar metrics, the authors propose an explainable human evaluation schema, collecting 20,000 annotations that include both standard ratings and fine-grained object/segment-level error tags. The study evaluates 14 models, revealing that closed-source systems generally lead, particularly in complex editing tasks and text-heavy domains, while also benchmarking VLM-based automated metrics against these human judgments.

**Strengths:**

- The benchmark addresses a clear gap in current evaluation by unifying a diverse set of tasks (generation and editing) and real-world domains (e.g., Information Graphics, Screenshots) under a single standardized protocol.

- The proposed explainable evaluation schema is a logical attempt to move beyond simple scalar scores. By asking annotators to explicitly tag object and segment-level failures, the benchmark provides richer diagnostic information on why models fail.

- The authors provide a large-scale study of 14 varied models, offering a comprehensive view of the current capabilities of both unified state-of-the-art systems and task-specific baselines.

**Weaknesses:**

- The automated evaluation hinges entirely on a proprietary model (Gemini-2.5-Flash) accurately assessing four distinct criteria (Relevance, Aesthetics, Coherence, Artifacts) and performing object listing. This reliance on a specific, closed-source model acts as a severe reproducibility bottleneck. If the model version changes or becomes unavailable, the automated portion of the benchmark is fundamentally compromised.

- The paper lacks necessary statistical rigor for a benchmark standard. It presents mean scores without conducting statistical significance tests. Without these, it is hard to determine if the performance gaps between top-performing models are meaningful or simply noise.

- While the paper reports inter-rater agreement (Krippendorff’s alpha), it fails to adequately address the high variability observed (ranging from a low 0.52 to 0.80 in Table 3). With such high disagreement for certain models/tasks, it is difficult to gauge the actual reliability of the human ground truth in those areas.

- The paper fails to cite or compare against established "decompose-and-verify" metrics that are highly relevant to fine-grained evaluation. A key missing reference is Gecko [1].


[1] Wiles et al., Revisiting text-to-image evaluation with gecko: On metrics, prompts, and human ratings, ICLR 2025

**Questions:**

- Can you clarify exactly how the VLM judge was implemented? Did it evaluate all four dimensions in a single pass? Have you considered that this might introduce a bottleneck, and have you experimented with specialized VLM judges for each distinct criterion to improve correlation?

- In Table 4, the VLM significantly under-penalizes artifacts compared to humans (positive bias). Can you provide some empirical evidence or analysis explaining this discrepancy? Is it a fundamental limitation of using current VLMs that may have been trained on noisy data?

- How was the prompt template for the VLM judge (Appendix A.7) derived? Did you utilize a "gold set" of human-annotated examples to validate and optimize this prompt before deployment?

- Regarding the reported Krippendorff’s alpha in Table 3, why was the least inter-annotator agreement found for the overall best-performing model (GPT-Image-1 in TIG)? What does this imply about the subjectivity of evaluating higher-quality outputs?

---

> ### Author Response · Authors · 2025-11-24
> **Author Response**
>
> We thank the reviewer for their insightful comments. Here, we give point-to-point responses to each concern or question.
> ***
> ***Weaknesses:***
> > Concern about reliance on a proprietary LLM (Gemini-2.5-Flash) for automated scoring.
>
> Our primary goal is to evaluate the ability of current large language models to serve as judges compared to human assessments. Therefore, the focus of our dataset is on human judgments rather than model-generated scores, and we collect human ratings accordingly. The LLM-based evaluation is included mainly for comparison purposes, not as the foundation of the benchmark. In addition, we cache all LLM-generated scores and will release both the human and LLM scores which should address the reproducibility concern.
>
>
> > Concern about insufficient statistical rigor.
>
> Thank you for highlighting the importance of statistical rigor. We appreciate the suggestion and have added comprehensive paired and independent t-tests in Appendix A.5 for all key comparisons. These analyses confirm that the observed performance differences are statistically significant and not attributable to random variation..
>
> > Concern about high inter-rater variability.
>
> Although the exact ranges vary, all values are above 0.5 (with only 2 out of 36 cases falling below 0.6). Similar ranges have been reported in prior work such as ImagenHub[1], suggesting that moderate variability is expected in open-ended image evaluation settings. In our dataset, we hypothesize this variability primarily reflects the presence of subjective subtopics (e.g., artworks, memes, posters) and the inclusion of subjective criteria, such as aesthetic quality, rather than inconsistencies in the annotation process. Moreover, both the mean trends and the standard deviations in human judgments closely align with the LLM-based scores, further supporting the reliability of our human annotations.
>
> > Missing comparison to “decompose-and-verify” metrics such as Gecko.
>
> Thank you for the suggestion. We have now added the citation of Gecko in the Related Work section. Gecko is indeed an important metric. However, we did not include Gecko in our experiments because it focuses on answering binary questions derived from the prompt, which primarily measures prompt relevance (only one of the four criteria in our human study). To keep human and LLM evaluations methodologically consistent, we follow the same scoring rubric for both.
> ***
> ***Questions:***
>
> > How was the VLM judge implemented? Did it evaluate all four dimensions in a single pass? ...
>
> As shown in Appendix A.7, after defining the task and scoring criteria, we ask the model to evaluate all four criteria in a single pass, since running separate evaluations for each would quadruple the computational cost. While specialized VLMs might improve correlation, we are not aware of VLMs specifically designed for our criteria, particularly for content coherence and artifact detection. We hope that our dataset can help address these limitations in future work.
>
> > Why does the VLM under-penalize artifacts?
>
> We provide two complementary analyses explaining why the VLM under-penalizes artifacts.
>
> **(1) scale misuse:** The VLM exhibits a bimodal scoring pattern. A total of 61.5% of outputs receive a score of 5, compared to 36.3% for humans, and it uses mid-range scores (3–4) much less often (13.8% vs. 45%). This collapse toward the extremes indicates that the model treats artifact detection almost as a binary task, which leads to systematic under-penalization of subtle defects.
>
> **(2) Domain bias from noisy and sparse training data:** Among 115 severe VLM failures (VLM = 5 and human ≤ 2), most occur in Screenshots (29.6%), Information Graphics (22.6%), and Computer Graphics (21.7%). These categories are dominated by noisy or low-quality examples in common web-scale pretraining datasets, while high-quality, well-rendered instances are relatively rare. As a result, the VLM develops weaker representations for these domains and is less sensitive to subtle artifacts.
>
>
> > How was the judge prompt template created?
>
> To ensure consistency between human and VLM evaluations, the prompt template follows the same format provided to our human annotators: we first define the task and then explain what each score from 1–5 means. Our primary goal in designing the prompt was to maintain consistency with human scoring guidelines rather than to optimize it using a gold set of annotated examples.
>
> > Why is Krippendorff’s alpha lower for GPT-Image-1?
>
> The lower agreement for GPT-Image-1 occurs because high-quality outputs leave fewer obvious mistakes for annotators to anchor on. When all images look good, judgments depend on subtle personal preferences, which increases subjectivity and reduces Krippendorff’s alpha. In contrast, weaker models produce clearer errors, leading to higher agreement.
> ***
> ***References:***
>
> [1] Ku et al .ImagenHub: Standardizing the evaluation of conditional image generation models, ICLR 2024

---

### Official Review · Reviewer_dy2F · 2025-11-01

**Soundness:** 3
**Presentation:** 3
**Contribution:** 3
**Rating:** 6
**Confidence:** 4

**Summary:**

The paper introduces ImagenWorld, a unified benchmark for image generation and editing that spans six task types (generation/editing with 0/1/multi references) and six topical domains, totaling 3.6K condition sets. It is supported by ~20K fine-grained human annotations and an explainable evaluation schema that tags object-level and segment-level issues, alongside automated VLM-based metrics. A large-scale study across 14 models reveals consistent trends: models struggle more with editing than generation, and with text-heavy/symbolic domains (e.g., screenshots, infographics), while targeted data curation can narrow gaps; VLM-as-judge correlates well with humans (Kendall up to 0.79) but lacks fine-grained error attribution. Overall, the work positions ImagenWorld as both a rigorous benchmark and a diagnostic tool for real-world image synthesis.

**Strengths:**

1. Substantially more “solid” than typical T2I-only benchmarks. Unlike prior works that center on narrow, compositional prompts, ImagenWorld covers six tasks × six domains with 3.6K condition sets and ~20K human annotations, and adds explainable labels (object-level and segment-level issue tags) that go beyond opaque scalar scores—this breadth and diagnostic depth should be highly valuable to the community.

2. Well-designed evaluation pipeline that bridges rigor and scalability. The study evaluates 14 diverse models under a single protocol and reports aligned human/VLM results (Kendall up to 0.79) while retaining human-annotated, localized error tags—striking a practical balance between scalable automatic scoring and explainable human diagnostics.

3. Clear, readable writing. The paper consistently defines evaluation criteria (Prompt Relevance, Aesthetic Quality, Content Coherence, Artifacts) and motivates them with concrete examples, which makes the methodology easy to follow.

**Weaknesses:**

The paper lacks a solution method showing how to leverage the benchmark’s diagnostics (e.g., a targeted data-curation or instruction-parsing baseline). Adding such a solution would make the benchmark more actionable with minimal extra work.

**Questions:**

Please see the weakness part.

---

> ### Author Response · Authors · 2025-11-24
> **Author Response**
>
> We thank the reviewer for their insightful comments on our paper. Below, we address the concern raised.
> ***
> ***Weaknesses and Questions:***
>
> > The paper lacks a solution method showing how to leverage the benchmark’s diagnostics (e.g., a targeted data-curation or instruction-parsing baseline). Adding such a solution would make the benchmark more actionable with minimal extra work.
>
> Since our dataset includes scores, object tags, and model outputs, it can be leveraged in three main ways: (1) as preference data, for example by training a model to imitate higher-rated outputs, avoid lower-rated ones, or rank outputs according to human preferences[1,2] (2) as diagnostic signals, for instance by using object tags to generate corrective instructions[3,4], and (3) as training data for developing explainable metrics that better align with human scores and capture object/segment-level issues in generated images[5,6]. We added a paragraph describing these possible use cases in the discussion section to address this concern.
> ***
> ***References:***
>
> [1] Wu, Xiaoshi, et al. "Human preference score: Better aligning text-to-image models with human preference." Proceedings of the IEEE/CVF International Conference on Computer Vision. 2023.
>
> [2] Liang, Youwei, et al. "Rich human feedback for text-to-image generation." Proceedings of the IEEE/CVF Conference on Computer Vision and Pattern Recognition. 2024.
>
> [3] Chakrabarty, Tuhin, et al. "Learning to follow object-centric image editing instructions faithfully." arXiv preprint arXiv:2310.19145 (2023).
>
> [4] Chen, Xinyan, et al. "Learning from mistakes: Iterative prompt relabeling for text-to-image diffusion model training." Findings of the Association for Computational Linguistics: EMNLP 2024. 2024.
>
> [5] Tan, Zhiyu, et al. "Evalalign: Supervised fine-tuning multimodal llms with human-aligned data for evaluating text-to-image models." arXiv preprint arXiv:2406.16562 (2024).
>
> [6] Kirstain, Yuval, et al. "Pick-a-pic: An open dataset of user preferences for text-to-image generation." Advances in neural information processing systems 36 (2023): 36652-36663.

---

> > ### Comment · Reviewer_dy2F · 2025-11-24
> >
> > Thanks for authors' rebuttal, I keep my positive rating.

---

> ### Author Response · Authors · 2025-11-26
>
> We thank Reviewer dy2F for the constructive feedback. We appreciate your careful reading and have addressed each point in the rebuttal. If any part remains unclear, we would be happy to clarify further. We kindly ask you to re-evaluate the paper in light of these revisions.

---

### Author Response · Authors · 2025-11-24
**ChangeLogs**

We thank all the reviewers for their thoughtful comments. We have considered each of the suggestions and have revised our manuscript accordingly. Here is the change log:

- As suggested by reviewer dy2F, we added a future direction for utilizing the benchmark in the Discussion section.
- As suggested by reviewer QiZo, we added the Gecko reference to the Related Work section, as well as a comparison to RICH-HF, which we found to be relevant to our work.
- As suggested by reviewer QiZo, we added a statistical significance analysis in Appendix A.5.
- As suggested by reviewer 7NZP, we revised the writing of Section 5.3 to clarify that the discussion is based on the four evaluation criteria.

---

### Author Response · Authors · 2025-12-03
**Comments to ACs/SACs**

Dear ACs,

Unfortunately, due to the new ICLR policies, the reviewers were not able to continue our discussion further. However, the reviews toward our paper were positive, and we tried to address the reviewers’ concerns as thoroughly as possible.

best regards,

---

### Meta-Review · Area_Chair_gRtV · 2025-12-13

**Summary:**

In short, the paper is generally viewed as solid and well-executed, but several concerns are:
1. Though fine-grained object- and segment-level annotations are a major strength, the paper does not sufficiently demonstrate how these diagnostics can be leveraged in practice.

2. The automated evaluation relies on a proprietary VLM, raising reproducibility concerns

**Reviewer Concerns:**

In general, reviewers address most of the concerns.

**Reviewer Scores:**

Two reviewers: 6: marginally above the acceptance threshold. But would not mind if paper is rejected
One reviwers: 8 accept

---

### Decision · Program_Chairs · 2026-01-26

Accept (Poster)